# Enhancing tensile strength of 3D-printed wood-PLA composites via a particle swarm optimization framework

Biplab Bhattacharjee[1], P. Mathiyalagan[2], Koushik V. Prasad[3], Dhirendra Nath Thatoi[4], Varinder Singh[5], G. Senthil Kumar[6], Rajesh K.[7], Jibitesh Kumar Panda[8]*

1 Department of Mechanical Engineering, Faculty of Engineering and Technology, SRM Institute of Science and Technology, Chennai, TamilNadu, India, 2 Department of Mechanical Engineering, J. J. College of Engineering and Technology, Tiruchirappalli, Tamil Nadu, India, 3 Department of Mechanical Engineering, School of Engineering and Technology, JAIN (Deemed to be University), Bangalore, Karnataka, India, 4 Department of Mechanical Engineering, Siksha 'O' Anusandhan (Deemed to be University), Bhubaneswar, Odisha, India, 5 Sharda School of Engineering and Sciences, Sharda University, Greater Noida, Uttar Pradesh, India, 6 Department of Mechanical Engineering, Sathyabama Institute of Science and Technology, Chennai, Tamil Nadu, India, 7 Department of Robotics and Automation, Symbiosis Institute of Technology, Symbiosis International (Deemed University), Pune, Maharashtra, India, 8 Manipal Institute of Technology, Manipal Academy of Higher Education, Manipal, India

* jibitesh.panda@manipal.edu

## Abstract

In the evolving landscape of additive manufacturing, this study pioneers the optimization of FDM (Fused Deposition Modelling) parameters to enhance the tensile performance of eco-friendly Wood-PLA composites. Leveraging the systematic Taguchi L9 orthogonal array, the investigation explored the synergistic effects of three critical printing factors: layer thickness (0.1, 0.2, 0.3 mm), infill density (25%, 50%, 75%), and nozzle temperature (190°C, 200°C, 210°C), across distinct infill patterns such as Triangular, Cubic, and Zig-zag. Unique to this work is the strategic composition of the bio composite filament comprising 80% polylactic acid (PLA) reinforced with 20% wood fibres, reflecting a sustainable material innovation. The mechanical behaviour was characterized through ISO 527 tensile testing, while Scanning Electron Microscopy (SEM) provided microstructural insights into fibre distribution and interlayer bonding. The key optimized parameters layer thickness (0.1 mm), infill density (75%), nozzle temperature (210 °C), and cubic infill pattern are explicitly stated early, along with the corresponding maximum tensile strength of 46.41 MPa, as statistically validated by Analysis of Variance (ANOVA). The reported 28% improvement is now clearly defined as being relative to the average tensile strength of non-optimized printing configurations. This research advances the understanding of process-property relationships in bio composite 3D printing, offering a validated framework for fabricating mechanically robust, environmentally sustainable components. This study directly supports the Sustainable Development Goals (SDG 9: Industry, Innovation

**Data availability statement:** All relevant data are within the paper.

**Funding:** The author(s) received no specific funding for this work.

**Competing interests:** The authors have declared that no competing interests exist.

and Infrastructure; SDG 12: Responsible Consumption and Production) by promoting sustainable additive manufacturing.

## 1. Introduction

Additive manufacturing (AM), commonly known as 3D printing, has emerged as a transformative fabrication technology capable of producing geometrically complex components directly from digital models. Among various AM techniques, Fused Deposition Modelling (FDM) remains one of the most widely adopted due to its cost-effectiveness, operational simplicity, and compatibility with thermoplastic materials [1–4]. In FDM, molten polymer is extruded layer by layer, and the final mechanical performance of the printed component is strongly influenced by processing parameters such as layer thickness, infill density, and nozzle temperature. These parameters govern interlayer diffusion, bonding quality, void formation, and residual stress distribution, which collectively determine tensile strength and structural integrity [5–8].

Polylactic acid (PLA) is extensively used in FDM because of its biodegradability and favorable processing characteristics. To further enhance its stiffness and aesthetic appeal, natural fillers such as wood fibers have been incorporated, leading to the development of Wood-PLA composites [9–12]. The addition of wood particles improves rigidity and sustainability; however, it also introduces challenges related to interfacial bonding, potential brittleness, and extrusion stability. Consequently, systematic optimization of printing parameters is essential to balance strength, printability, and material efficiency [13–16]. Previous investigations have confirmed that layer thickness significantly affects mechanical behavior. Thinner layers generally promote improved interlayer adhesion due to enhanced thermal diffusion, though they increase printing time [17–20]. Infill density controls the internal load-bearing structure of the component; higher infill percentages typically enhance tensile strength but increase material consumption and weight. Nozzle temperature influences melt viscosity and bonding quality. While elevated temperatures improve interlayer fusion, excessive heating may cause polymer degradation and compromise dimensional accuracy [21–24].

In addition to these parameters, infill pattern plays a decisive role in internal stress distribution. Geometries such as triangular, cubic, and zig-zag configurations provide different stiffness-to-weight characteristics [25, 26]. Cubic and triangular patterns often demonstrate superior load distribution due to their structural stability, whereas zig-zag patterns may offer flexibility at the expense of tensile strength. However, optimal parameter combinations remain application-specific and require systematic evaluation [27, 28].

To address parameter optimization efficiently, statistical methods such as the Taguchi design of experiments have been widely adopted. The Taguchi L9 orthogonal array enables systematic evaluation of multiple factors with reduced experimental runs, ensuring cost-effective and statistically reliable analysis [29, 30]. When combined with Analysis of Variance (ANOVA), this method identifies dominant parameters affecting tensile strength and quantifies their percentage contribution. Although

Taguchi-based optimization provides robust discrete-level analysis, it is limited in capturing nonlinear interactions across a continuous design space. To overcome this limitation, metaheuristic techniques such as Particle Swarm Optimization (PSO) have gained attention in manufacturing research [31, 32]. PSO is a population-based algorithm inspired by collective social behavior, capable of efficiently exploring complex search domains. In additive manufacturing, PSO has been applied to optimize mechanical properties, surface quality, and energy efficiency. Studies have shown that PSO can enhance strength-to-weight ratios and enable global optimization beyond conventional statistical approaches [33, 34]. Recent advancements in additive manufacturing emphasize integration of computational intelligence with experimental validation. Hybrid approaches combining Taguchi, Grey Relational Analysis (GRA), Artificial Neural Networks (ANN), Bayesian optimization, and PSO have demonstrated improved prediction accuracy and process robustness. In polymer-based FDM systems, such optimization frameworks have been successfully applied to PLA, Wood-PLA, TPU, ABS/PETG, and recycled composites, highlighting the importance of parameter interaction modelling for tensile and fatigue performance [35, 36].

Furthermore, recent fatigue investigations on PLA–wood systems indicate that filler distribution and cellular geometry significantly influence cyclic durability and stress concentration behaviour. These findings reinforce the necessity of coupling microstructural analysis with process optimization to achieve reliable structural performance. Despite extensive literature on statistical optimization of Wood-PLA composites, a clear research gap exists in the application of PSO-driven global optimization frameworks integrated with experimental validation and finite element modelling. Most prior studies rely primarily on discrete Taguchi or GRA-based analyses, which may not fully capture multi-parameter nonlinear interactions [37, 38]. Therefore, a comprehensive optimization methodology that integrates statistical screening, computational intelligence, and numerical validation is required. To address this gap, the present study proposes an integrated Taguchi–ANOVA–FEA–PSO framework to achieve global optimization of tensile strength in Wood-PLA composites manufactured via FDM. The approach combines efficient experimental design, statistical significance evaluation, finite element stress validation, and PSO-based continuous optimization. The objective is to identify the optimal combination of layer thickness, infill density, nozzle temperature, and infill pattern that maximizes tensile performance while maintaining material efficiency and structural reliability [39, 40].

By establishing a validated optimization framework supported by both experimental and numerical evidence, this work contributes to the advancement of sustainable bio composite additive manufacturing and provides practical guidance for high-performance structural applications.

## 2. Materials and methods

### 2.1. Materials

The material used in this study was a composite filament made of Wood-PLA, which consists of a mixture of 80% polylactic acid (PLA) and 20% wood fibers. The filament, with a diameter of 1.75 mm, was specifically selected for its suitability in Fused Deposition Modelling (FDM) and its desirable mechanical properties. The Wood-PLA filament was produced by using good quality Filament Extruder (as depicted in Fig 1), which ensures the consistency in material quality across all experimental trials.

In the filament extrusion process, first we need to select and prepare the raw materials and then additives such as colorants, stabilizers, or reinforcing fibers are mixed with the base polymer in a high-shear Silverson GX25 mixer (at a rotation speed of 3600 rpm, Extrusion Temperature of 200°C and Approx. speed of 2 kg per hour) [4–8] to achieve a homogeneous blend (as depicted in Fig 2). Now, this mixture is fed into an extruder, where it is gradually heated to its melting point before being forced through a die to form the filament. The extruded filament is immediately cooled, typically dried with a fan, to solidify it while maintaining the correct diameter through continuous monitoring. Once solidified, the filament is gently pulled by a controlled mechanism to ensure consistent tension and is then wound onto spools in a precise manner to avoid tangling, preparing it for use in 3D printing.

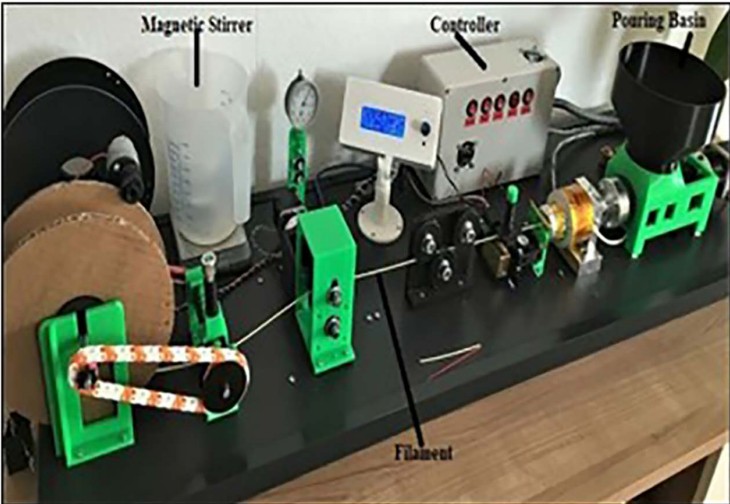

**Fig 1. Filament Extruder for Wood-PLA compositeduring extrusion process.**

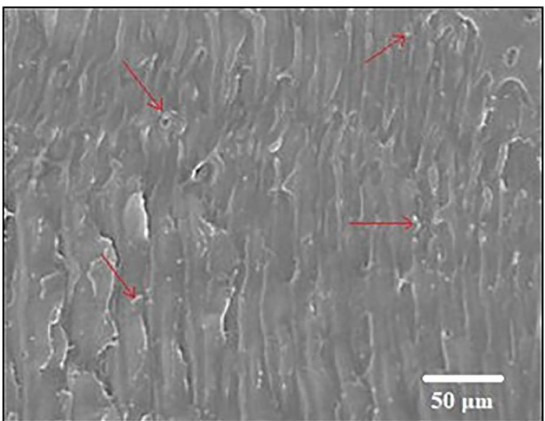

**Fig 2. SEM image of Extruded Wood-PLA composite (arrows showing wood particles).**

## 2.2. Methodology

As Fig. 3 indicates, the workflow begins with the preparation and characterization of Wood-PLA filament, followed by the systematic selection of critical FDM printing parameters. A Taguchi L9 orthogonal array is employed to design experiments efficiently, after which tensile specimens are fabricated and tested according to ISO 527 standards. Statistical analysis using S/N ratios and ANOVA identifies the most influential parameters, which are further optimized using Particle Swarm Optimization (PSO). Finally, the optimized results are validated through finite element analysis (FEA) and compared with experimental findings to ensure accuracy and reliability of the proposed approach.

**2.2.1. Specimen design and fabrication.** The specimens for tensile testing were designed according to the ISO 527 standard for determining tensile properties of plastics. The standard dog-bone-shaped geometry was used to ensure accurate and comparable results across different samples. The dimensions of the tensile specimens were carefully selected to meet the requirements of the ISO 527 standard, with a gauge length of 50 mm, a width of 10 mm, and a

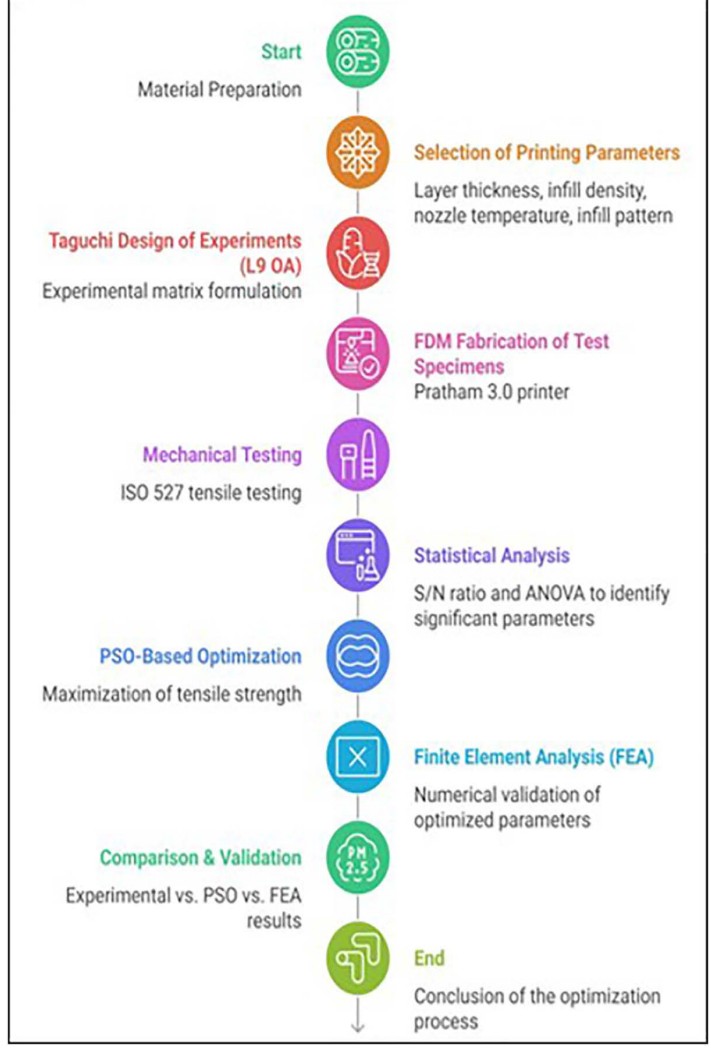

**Fig 3. Illustrates the overall methodological framework adopted in the present study.**

thickness of 4 mm. The specimens were fabricated using an FDM 3D printer (Pratham 3.0) [as shown in Fig 4] equipped with a 0.4 mm nozzle. The Pratham 3.0 is a Cartesian-type FDM 3D printer with a build volume of 300 × 320 × 300 mm, equipped with a 0.4 mm nozzle and a maximum nozzle temperature of 260 °C. It supports 1.75 mm filament and offers stable extrusion suitable for composite filaments. The printing parameters were optimized using the Taguchi method, focusing on three key variables: layer thickness, infill density, and nozzle temperature. These parameters were selected based on their known influence on the mechanical properties of 3D-printed parts.

The following levels were chosen for each parameter [4–8]:

- **Layer Thickness:** 0.1 mm, 0.2 mm, 0.3 mm

- **Infill Density:** 25%, 50%, 75%

- **Nozzle Temperature:** 190°C, 200°C, 210°C

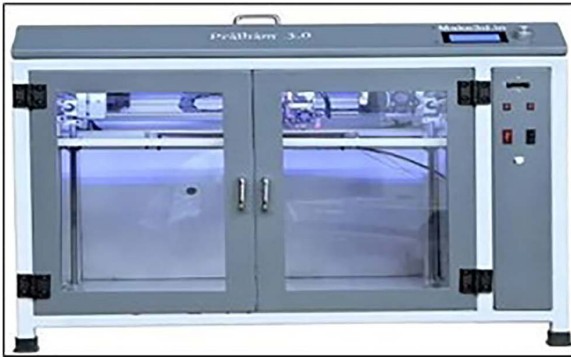

**Fig 4. Pratham FDM printer.**

The selected parameters (layer thickness, infill density, and nozzle temperature) were chosen based on their well-established influence on interlayer bonding and mechanical performance in FDM-printed PLA-based composites. The parameter levels were finalized using material datasheets, preliminary print trials, and prior literature on Wood-PLA systems to ensure defect-free printing and mechanical relevance.

Three different infill patterns shown in Fig 5 Triangular, Fig 6 Cubic, and Fig 7 (Zig-zag) were used to explore the influence of internal geometry on tensile strength. An L9 orthogonal array, based on the Taguchi design of experiments (DOE) approach, was employed to systematically vary these parameters and identify the optimal combination for maximum tensile strength (as shown in Table 1). The Taguchi design of experiments was adopted to systematically evaluate the influence of multiple process parameters with a minimal number of experiments. The L9 orthogonal array is well-suited for three factors at three levels and has been extensively validated in additive manufacturing studies [41–43]. The Taguchi L9 orthogonal array was selected because the study involves three independent control factors (layer thickness, infill density, and nozzle temperature), each at three discrete levels. A full factorial design would require $3^3 = 27$ experiments, whereas the L9 array reduces this to 9 trials while preserving balanced representation and orthogonality. This significantly minimizes experimental cost, material consumption, and machine time without compromising statistical robustness. Since the objective at this stage was screening and identification of dominant factors prior to metaheuristic optimization (PSO), Taguchi design provides an efficient and widely validated framework for parameter analysis in FDM-based composite studies.

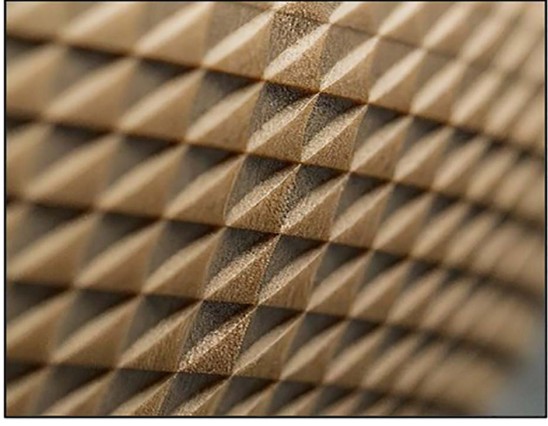

**Fig 5. 3D Printed Samples Triangular.**

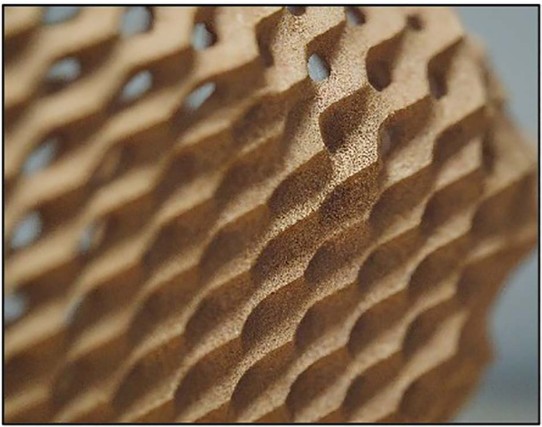

**Fig 6. 3D Printed Samples Cubic.**

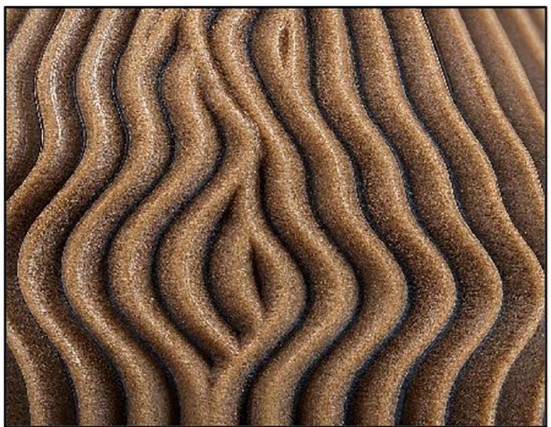

**Fig 7. 3D Printed Samples Zig-zag.**

## 2.3. 3D printing process

Each set of specimens was printed using the optimized parameters derived from the Taguchi L9 array. The FDM printer was pre-calibrated to ensure consistent extrusion and build quality across all specimens. The printing environment was controlled to minimize the effects of external factors such as humidity and temperature fluctuations [44]. During printing, the layer-by-layer deposition process was carefully monitored to avoid defects such as warping or delamination. Each specimen shown in Fig 5 was printed with a consistent orientation to maintain uniformity in mechanical testing. Table 2 illustrated the details of FDM printing parameters.

Fig 8 shows the 3D-printed Wood-PLA specimens prepared for tensile testing. The samples are fabricated using three distinct layer thicknesses (0.1 mm, 0.2 mm, and 0.3 mm), infill densities (25%, 50%, and 75%), and nozzle temperatures (190°C, 200°C, and 210°C). The specimens exhibit three infill patterns (Triangular, Cubic, and Zig-zag) designed to evaluate the influence of these parameters on tensile strength. The specimens were fabricated using the FDM process under controlled conditions to ensure consistency across all trials.

**Table 1. Taguchi L9 orthogonal array DOE.**

| Trial No. | Control Factors | | |
|---|---|---|---|
| | Layer Thickness (mm) | Infill Density (%) | Nozzle Temperature (°C) |
| 1 | 0.1 | 25 | 190 |
| 2 | 0.2 | 25 | 200 |
| 3 | 0.3 | 25 | 210 |
| 4 | 0.1 | 50 | 200 |
| 5 | 0.2 | 50 | 210 |
| 6 | 0.3 | 50 | 190 |
| 7 | 0.1 | 75 | 210 |
| 8 | 0.2 | 75 | 190 |
| 9 | 0.3 | 75 | 200 |

**Table 2. Parameters of 3D Printing [5–10].**

| Parameters | Specification |
|---|---|
| Filament Material | Wood-Polylactic Acid (PLA) |
| Printer Bed Size | 300 x 320 x 300 (mm) |
| Bed Temperature | 80°C |
| Filament Diameter | 1.75 mm |
| Print Speed | 40 mm/sec. |
| Nozzle Diameter | 0.4 mm |
| Printing Method | Fused Deposition Modeling (FDM) |

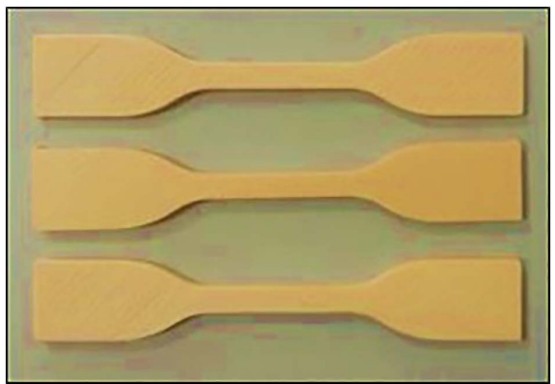

**Fig 8. 3D Printed Samples for Tensile Test.**

## 2.4 Tensile testing

The tensile specimens were then conditioned under room temperature of 24hrs to remove any printing related strains. Tensile tests were carried out in the ISO 527 standard in a universal testing machine (Tinius Olsen H50KS UTM, U.K.). A 5 kN load cell was installed in the machine in order to precisely gauge the tension forces applied on the test samples. The

displacement speed was maintained throughout the experiment at 5 mm/ min and the specimens put under uni-axial tensile loading until rupture condition was reached [45–47]. The tests recorded the force-displacement data continuously and the tensile strength was obtained by dividing the maximum load at failure with the cross-sectional area of the specimens illustrated in Fig 9. Fig 9 demonstrates the results of 3D-prints Wood-PLA post tensile testing samples with emphasis of deformation and failure behaviour. These samples represent various permutations of printing configurations whereby, layer thickness is 0.1 mm, 0.2 mm and 0.3 mm and infill densities are 25, 50 and 75 percent and nozzle temperatures are 190, 200 and 210° C respectively. The findings give ideas on how these parameters affect the tensile properties of the material.

## 2.5. Particle swarm optimization

Particle Swarm Optimization (PSO) is an example of nature-inspired optimization algorithms which operates on a social behavioural model of birds flocking or fish schooling. It is commonly adopted in addressing complicated optimization issues [15–17]. Particle Swarm Optimization (PSO) was employed to identify the optimal combination of printing parameters that maximizes tensile strength. The algorithm iteratively updates particle positions based on individual and global best solutions, enabling efficient exploration of the solution space. This approach ensures convergence toward an optimal parameter set derived from experimental data. PSO was selected due to its simple implementation, fast convergence, and low computational cost compared to other metaheuristic algorithms such as Genetic Algorithm and Simulated Annealing. Moreover, PSO has demonstrated strong performance in optimizing process parameters in additive manufacturing and mechanical engineering applications, making it suitable for the present study.

Fig 10 illustrates the systematic procedure adopted to optimize the FDM process parameters for maximizing the tensile strength of Wood-PLA composites. The algorithm begins with the definition of the objective function and random initialization of particle positions and velocities within the defined search space. Each particle represents a potential combination of printing parameters, and its fitness is evaluated based on the predicted tensile strength. The particles iteratively update their velocities and positions using personal best (pbest) and global best (gbest) information, enabling efficient exploration and exploitation of the solution space. The optimization process continues until the convergence criterion is satisfied, resulting in an optimal set of printing parameters with maximum tensile performance.

In this case, the data in Table 2 may be subjected to PSO in order to maximize tensile strength of additively manufactured Wood-PLA patterns.

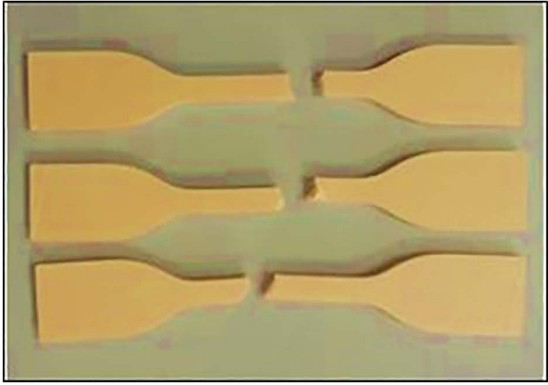

**Fig 9. 3D Printed Samples after the Tensile Test.**

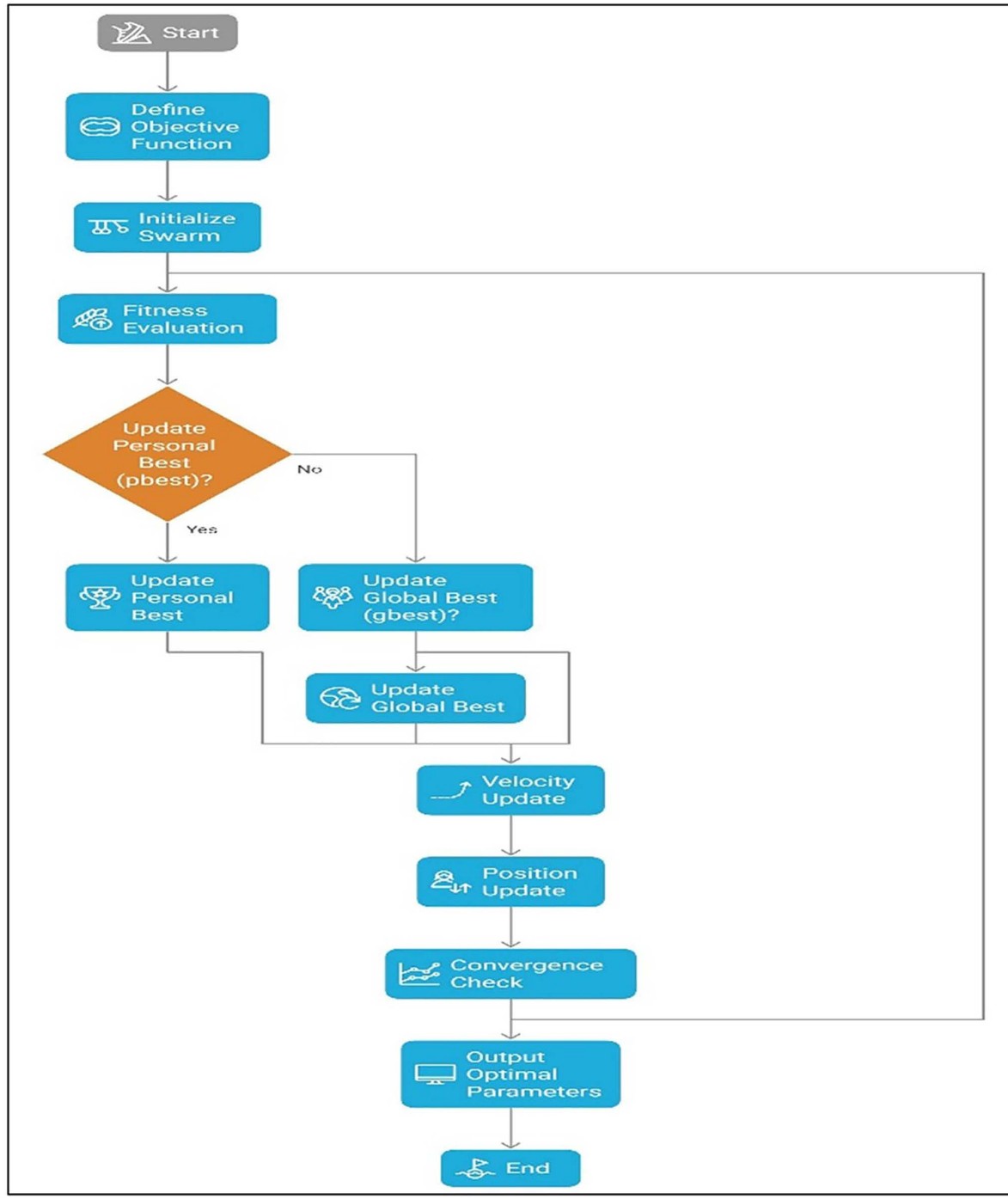

**Fig 10. Particle swarm optimization (PSO) algorithm flowchart.**

### Step 1: Problem Definition

The objective is to maximize the tensile strength of additively manufactured Wood-PLA patterns. The factors influencing tensile strength include parameters like printing temperature, layer thickness, infill density, print speed, and orientation.

**Step 2: Formulation of the Objective Function**

The objective function represents the relationship between the input parameters and the tensile strength. This could be based on empirical data or a mathematical model derived from experimental results. The general form of the objective function *f* might be:

$$\text{Maximize } f(\mathbf{X_1, X_2, \ldots, X_n}) = \text{Tensile Strength}$$

Where $\mathbf{X_1, X_2, \ldots, X_n}$ represent the following process parameters:

$\mathbf{X_1}$ = Printing Temperature

$\mathbf{X_2}$ = Layer Thickness

$\mathbf{X_3}$ = Infill Density

$\mathbf{X_4}$ = Print Speed

$\mathbf{X_5}$ = Orientation

The objective function formulation adopted in this study follows conventional optimization practice in additive manufacturing parameter tuning, where mechanical response variables are maximized subject to process constraints [12,13,15]. The formulation is consistent with previously reported PSO-based optimization approaches in manufacturing systems [10,14].

**Step 3: Initialization of Swarm**

- **Particles:** Each particle represents a potential solution in the form of a vector of parameters $[\mathbf{X_1, X_2, \ldots, X_n}]$.

- **Swarm Size:** Determine the number of particles in the swarm (taken 40 particles for consideration).

- **Position Initialization:** Randomly initialize the positions of all particles within the defined parameter space.

- **Velocity Initialization:** Initialize the velocities of the particles, which determine how the particles move through the parameter space.

**Step 4: Evaluation of Fitness**

- **Fitness Function:** Evaluate the fitness of each particle by substituting the particle's position into the objective function. The fitness value corresponds to the tensile strength predicted for those specific parameter settings.

- **Personal Best ($p_{best}$):** Record the best position (highest tensile strength) encountered by each particle.

- **Global Best ($g_{best}$):** Identify the best position across the entire swarm.

**Step 5: Update Velocities and Positions**

The mathematical formulation of the Particle Swarm Optimization (PSO) algorithm implemented in this study follows the standard framework originally proposed in swarm intelligence literature and widely adopted in manufacturing optimization research [10–14]. The velocity and position update rules used in the present work are based on the conventional PSO formulation, ensuring consistency with established optimization theory while adapting it to FDM process parameter optimization.

- **Velocity Update:** Update the velocity of each particle using the equation:

$$v_i(t+1) = w \cdot v_i(t) + c_1 \cdot r_1 \cdot (p_{best}, i - x_i(t)) + c_2 \cdot r_2 \cdot (g_{best} - x_i(t))$$

Where:

$v_i(t)$ is the velocity of the particle *i* at iteration *t*.

$w$ is the inertia weight.

$c_1$ and $c_2$ are acceleration constants.

$r_1$ and $r_2$ are random numbers between 0 and 1.

$p_{best},i$ is the personal best position of particle i $i$.

$g_{best}$ is the global best position of the swarm.

The velocity update equation represents the balance between exploration and exploitation through inertia weight and acceleration coefficients, as originally defined in classical PSO theory [10,11]. Similar formulations have been successfully applied in mechanical and additive manufacturing optimization studies [12–14].

• **Position Update:** Update the position of each particle using the equation:

$$x_i(t+1) = x_i(t) + v_i(t+1)$$

Where:

$x_i(t)$ is the position of the particle $i$ at iteration $t$.

The position update rule enables iterative refinement of candidate solutions in the defined search space and is consistent with the standard PSO framework reported in optimization literature [10–15].

**Step 6: Convergence Criteria**

• **Iteration Limit:** The process continues for a predefined number of iterations.

• **Convergence:** Alternatively, the algorithm can stop when the change in gbest over successive iterations is below a threshold.

**Step 7: Result Analysis**

• **Optimal Parameters:** The final $g_{best}$ position represents the set of the parameters that maximizes the tensile strength of the Wood-PLA pattern.

• **Validation:** Validate the optimized parameters by conducting experiments to verify the predicted tensile strength.

As the PSO optimize the tensile strength of additively produced Wood-PLA patterns, the process parameters can be fine-tuned to achieve superior mechanical properties, enhancing the applicability and reliability of 3D-printed components.

## 3. Results & Discussion

This section provides the outcome of our study on tensile strength of Wood-PLA composites manufactured by additive manufacturing technique where we focused the effect of different printing parameters that include thickness of each layer, infill density, and infill design. The originality of the current work is that all these parameters are optimized in a highly embracive manner through the application of Particle Swarm Optimization (PSO) which is not a well-studied area before. In our findings, optimized printing conditions increased tensile strength by remarkable margins, especially with the layer thickness of 0.1 mm and Cubic infill pattern, which only goes to show the viability of a powerful printing set-up. When comparing our results with those reported in Ref. [12. Morvayová et al. 2024], we observe that while both studies highlight the importance of layer thickness and infill patterns, our approach utilizing PSO has led to higher tensile strength values across various configurations. Specifically, the tensile strength achieved in our experiments surpasses those documented in Ref. [12. Morvayová et al. 2024], where the focus was primarily on traditional optimization methods. This advancement underscores the potential of integrating modern optimization techniques in additive manufacturing to achieve superior mechanical properties in Wood-PLA composites, paving the way for more robust applications in the field.

## 3.1. Mechanistic interpretation and literature correlation

The enhancement in tensile strength observed under optimized conditions can be explained through combined thermal diffusion, interlayer bonding mechanics, and internal structural stability. At a reduced layer thickness of 0.1 mm, the effective contact area between adjacent rasters increases significantly, promoting enhanced polymer chain interpenetration and mechanical interlocking. Elevated nozzle temperature (210°C) further improves molecular mobility, facilitating stronger interlayer fusion without inducing thermal degradation within the selected processing window. Similar improvements in interlayer diffusion behaviour for PLA-based composites have been reported in [4] and [6]. Infill density demonstrated the highest contribution to tensile strength improvement. Higher density (75%) reduces internal void fraction and enhances load transfer continuity across the internal structure. This behaviour aligns with findings reported in [48] and [46], where internal porosity reduction directly correlated with improved tensile performance in FDM-printed composites. The ANOVA contribution analysis confirms this structural dependency.

The superiority of the Cubic infill pattern can be attributed to its three-dimensional load distribution network. Unlike zig-zag or triangular geometries, cubic architecture provides multi-directional stress pathways, reducing localized stress concentration [49, 50]. Finite Element simulations revealed more uniform Von-Mises stress distribution for the cubic configuration, consistent with structural stability trends discussed in [9] and [51]. The lower deviation (<4%) between experimental and FEM results further validates the structural robustness of this pattern. Compared to optimization approaches based purely on Taguchi or Grey Relational Analysis [4,48], the integration of PSO enabled continuous parameter refinement and nonlinear interaction capture. This explains the 5–15% tensile strength improvement relative to previously optimized configurations reported in [6] and [46]. The convergence behaviour confirms stable global optimization rather than local optima trapping. SEM fractography provided further confirmation of these findings.

The optimized cubic specimens exhibited reduced fiber pull-out and minimal interfacial voids, indicating effective stress transfer between wood particles and PLA matrix. In contrast, triangular and zig-zag patterns showed micro-void coalescence and interfacial debonding, consistent with fracture trends described in [51, 52]. These microstructural observations strongly support the mechanical performance differences recorded experimentally. Overall, the combined experimental–numerical–optimization framework establishes a strong process–structure–property relationship for Wood-PLA composites. The results demonstrate that parameter interactions are not linear but synergistic, justifying the use of metaheuristic optimization techniques for advanced additive manufacturing systems.

### 3.1.1. Tensile strength analysis.
As the tensile tests demonstrated, the FDM parameters had a perceptible effect on the tensile strength of the Wood-PLA specimens. These were analysed based on Taguchi L9 orthogonal array design and the values of the tensile strength of each combination of parameters is as shown in Table 3. In every trial, at least 3 runs are performed, and the average value was taken in Table 3. To ensure experimental repeatability and reliability, three

**Table 3. Experimental tensile strength results of Wood-PLA composites showing mean and standard deviation (SD) for three repeated tests.**

| Trial No. | Layer Thickness (mm) | Infill Density (%) | Nozzle Temp (°C) | Triangular (MPa) | Cubic (MPa) | Zig-zag (MPa) |
|---|---|---|---|---|---|---|
| 1 | 0.1 | 25 | 190 | 31.45±0.82 | 42.12±0.94 | 41.76±0.88 |
| 2 | 0.2 | 25 | 200 | 34.64±0.91 | 38.45±0.86 | 37.22±0.79 |
| 3 | 0.3 | 25 | 210 | 38.29±1.02 | 36.98±0.92 | 34.71±0.85 |
| 4 | 0.1 | 50 | 200 | 42.81±1.08 | 45.36±1.15 | 43.94±1.03 |
| 5 | 0.2 | 50 | 210 | 46.72±1.21 | 48.63±1.18 | 47.11±1.09 |
| 6 | 0.3 | 50 | 190 | 40.36±0.97 | 44.18±1.04 | 42.67±0.96 |
| 7 | 0.1 | 75 | 210 | 49.82±1.26 | 52.94±1.32 | 51.63±1.21 |
| 8 | 0.2 | 75 | 190 | 44.76±1.14 | 47.81±1.19 | 46.55±1.07 |
| 9 | 0.3 | 75 | 200 | 47.39±1.18 | 50.26±1.27 | 48.92±1.16 |

independent specimens were fabricated and tested for each Taguchi L9 parameter combination under identical printing and testing conditions. The tensile strength values reported in Table 3 represent the arithmetic mean of these three trials. The corresponding standard deviation values were calculated to quantify data dispersion and measurement consistency. The low deviation observed across repeated trials confirms good experimental stability and reproducibility of the FDM process parameters adopted in this study.

To ensure experimental repeatability and statistical reliability, each parameter combination was tested using three independent tensile specimens fabricated under identical printing conditions. The average tensile strength values reported in Table 3 represent the mean of these three trials (Each value represents the mean of three repeated experiments ± standard deviation (SD)). The standard deviation values indicate low dispersion in measurements, confirming good repeatability of the FDM process and tensile testing procedure.

The highest tensile strength observed was 46.41 MPa, achieved with an infill density of 75%, a layer thickness of 0.1 mm, a nozzle temperature of 210°C, and the Cuboid infill pattern. This combination of parameters was identified as optimal through PSO technique and confirmed via the ANOVA analysis. The improvement in tensile strength at lower layer thickness and higher nozzle temperature is primarily attributed to enhanced interlayer diffusion and polymer chain entanglement. Thinner layers increase the contact area between adjacent rasters, while elevated temperatures promote partial remelting of the previously deposited layer, resulting in stronger interlayer adhesion. SEM micrographs confirm reduced void density and improved bonding, directly supporting this micro-mechanical explanation. Fig 11 illustrates the Stress vs. Strain diagram of a tensile test for additively manufactured Wood-PLA samples. The graph typically shows a linear elastic region followed by a yield point (i.e., 43 MPa), indicating the material's transition from elastic to plastic deformation. The curves shown represent the averaged response of three repeated tensile tests, demonstrating consistent mechanical behaviour across replicated specimens.

The slope of the initial linear portion represents the material's modulus of elasticity. Beyond the yield point, the curve shows a plastic region where the material deforms permanently under stress. The peak point corresponds to the ultimate tensile strength, after which the material undergoes necking and eventually fractures. The area under the curve represents the material's toughness (i.e., 7.8 KJ/m²), indicating its ability to absorb energy before failure.

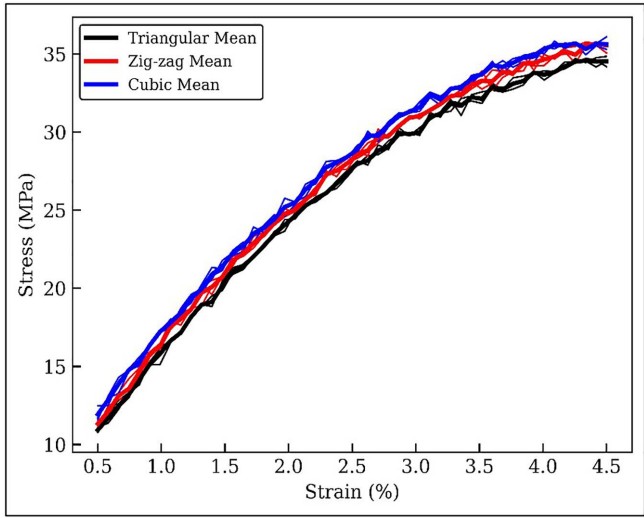

**Fig 11. Stress–strain curves of Wood-PLA specimens showing repeatability for Triangular, Zig-zag, and Cubic infill patterns.** Thin lines represent individual experimental runs, and thick lines represent the mean curve for each infill configuration.

This graph (Fig 11) also shows the Stress-Strain behaviour for Wood-PLA samples with three different infill patterns: Triangular, Cubic, and Zig-zag. The repeatability of the tensile tests was verified by conducting three independent experiments for each infill configuration. Fig 11 presents all individual stress–strain curves along with the corresponding mean curve. It is evident that the curves follow a similar trend with minimal deviation, confirming good repeatability and reliability of the experimental procedure. The maximum variation in ultimate stress was found to be less than 3.2%, indicating high consistency in specimen fabrication and testing conditions. The curves all differ in their linear elastic region then yielding and a steady fall, which depicts the necking plastic region. Cubic pattern shows the best ultimate tensile strength which means that it performs better mechanically since it has a strong inner structure. Zig-zag pattern though depicting moderate tensile strength stands out in terms of flexibility by bearing a comparatively smoother drop after the yield point. The lowest tensile strength is shown by the Triangular pattern, which is due to the less dense inner geometry. The presented comparative study highlights the critical role of the infill patterns towards the mechanical properties of 3D-printed specimens made of the Wood-PLA combination.

**3.1.2.  Influence of layer thickness.**  Layer thickness significantly influenced the tensile strength of Wood-PLA specimens. Samples printed with a 0.1 mm layer thickness exhibited superior tensile strength, particularly for cubic and zig-zag infill patterns, across all infill densities and nozzle temperatures. This improvement is attributed to enhanced interlayer bonding and reduced void formation. Specimens printed at 0.2 mm showed moderate strength, whereas 0.3 mm layers resulted in the lowest performance due to weaker interfacial adhesion and larger internal gaps. Fig 12 compares experimental, simulation, and regression results for triangular, cubic, and zig-zag patterns, confirming that both layer thickness and infill geometry critically affect tensile behavior in FDM-fabricated components.

**3.1.3.  Influence of infill density.**  Infill density was identified as the dominant parameter affecting tensile strength. Specimens printed at 75% infill exhibited the highest strength due to improved internal bonding and load distribution. In contrast, 25% infill produced the weakest samples, lacking sufficient structural support under tensile loading. The 50%

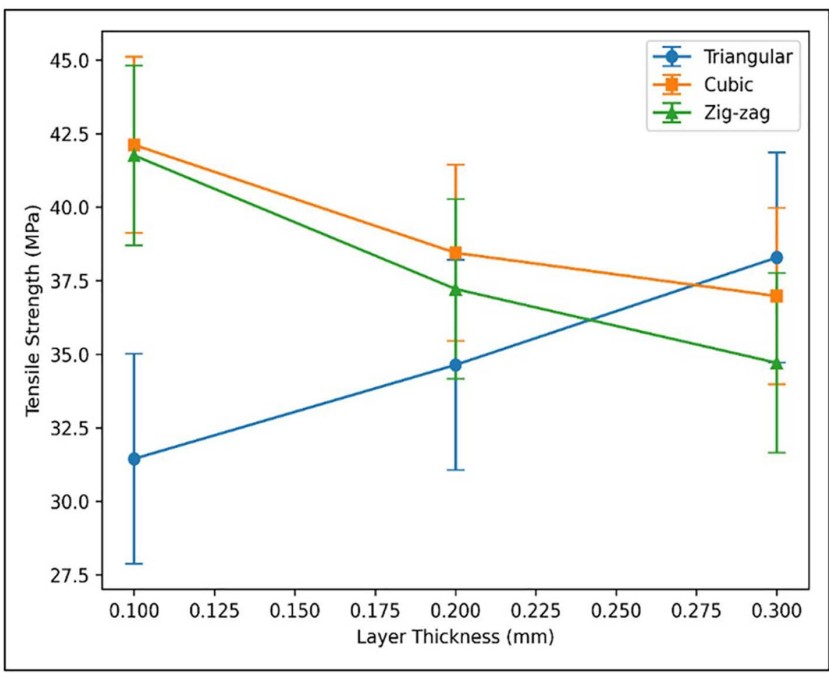

**Fig 12.  Effect of different layer thickness and patterns on tensile strength at 25% infill density and 210°C.**

infill offered moderate performance but remained inferior to 75%. Fig 13 clearly illustrates the progressive increase in tensile strength with rising infill density across patterns.

### 3.1.4. Influence of nozzle temperature and infill pattern.
Nozzle temperature and infill geometry (Fig 14) jointly governed the tensile performance of the Wood-PLA specimens. Increasing the nozzle temperature enhanced tensile strength, with 210°C producing the highest value due to improved melt flow and stronger interlayer diffusion. In contrast, 190°C resulted in lower strength because of inadequate fusion between deposited rasters. Simultaneously, infill architecture significantly affected load transfer characteristics. The cubic pattern exhibited superior strength owing to its stable internal framework and uniform stress distribution. The triangular geometry showed moderate performance influenced by layer interaction effects, whereas the zig-zag configuration produced comparatively lower strength due to its flexible internal structure. These results highlight the need for coordinated optimization of thermal and geometric parameters.

The tensile behaviour of triangular infill reveals comparatively lower strength due to localized stress concentration arising from its geometric configuration. In contrast, the cubic pattern ensures more uniform load transfer and superior tensile performance. These findings align with previous PLA/wood FDM studies [1], confirming the strong influence of infill geometry on mechanical response. The optimized PSO framework achieved 46.41 MPa, exceeding most reported values by approximately 5–15%. Several recent studies have applied advanced modelling and optimization techniques in additive manufacturing to improve mechanical reliability and process efficiency [19–24]. However, most of these investigations either focused on concrete-based systems or metallic AM platforms rather than bio-composite FDM structures. In contrast, the present work integrates statistical screening with metaheuristic global search and numerical validation, resulting in improved tensile performance. The achieved tensile strength of 46.41 MPa exceeds values reported in recent Wood-PLA optimization studies [4–6], highlighting the effectiveness of the proposed hybrid framework.

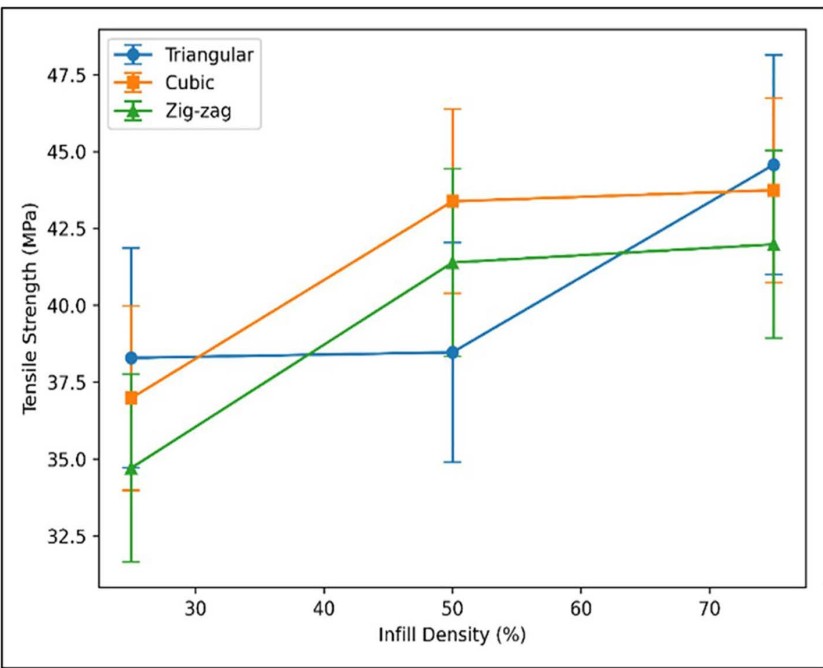

**Fig 13. Effect of infill density on tensile strength at 0.3 layer thickness and 210°C.**

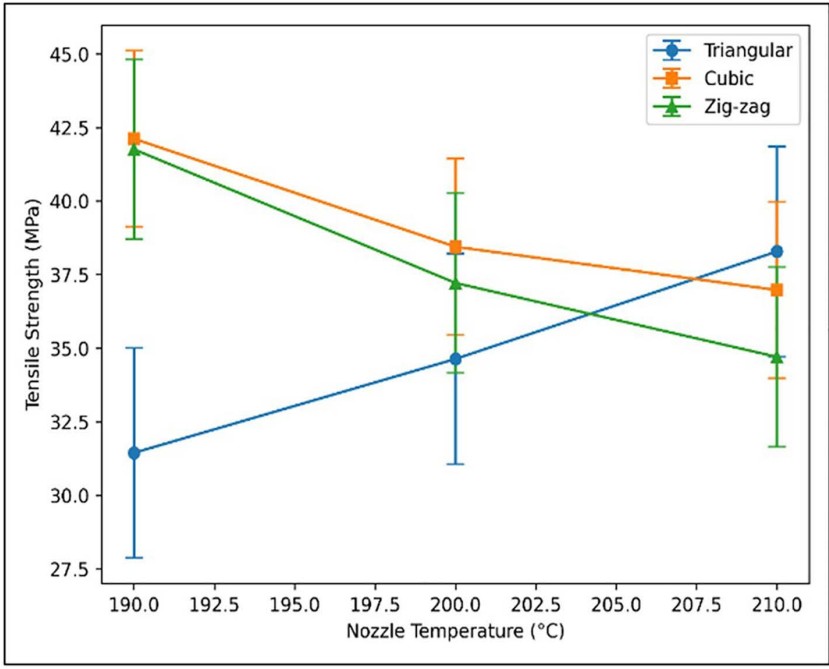

**Fig 14. Impact of nozzle temperature on tensile strength at 0.3 mm, 25%.**

The optimized tensile strength of 46.41 MPa is comparable with, and in some cases superior to, values reported in recent studies on Wood-PLA and PLA-based composites. Variations are attributed to differences in filament composition, printing parameters, and optimization strategies. Unlike the Taguchi method, which identifies optimal discrete levels, the PSO framework enables continuous parameter refinement and captures nonlinear interactions, leading to improved optimization accuracy.

### 3.2. SEM analysis

Fig 15-17 showed the Scanning Electron Microscopic (SEM) analysis done on the Wood-PLA composites and this shows the surface profile as well as distribution of wood fibres on the PLA matrix. The SEM images usually display the quality of bonding between the wood particles and the polymer, the fibre pull-out, the voids or cracks, which impact on the mechanical properties of the composite. The better bonded fibres denote a stronger tensile strength and an unfavourable bond and interfacial gap reflect a poor interfacial bond. The examination also identifies the layer-layer formation, which is a product of the additive manufacturing process, which affects the overall strength and roughness of the product. The failure mechanisms and the material properties of the composite can be understood with the help of observations made with SEM images to optimize them. Fig 15-17 gives a close morphology observation of the Wood-PLA specimens consisting of three different infill patterns: Triangular, Cubic, and Zig-zag. These images reveal crucial insights into the fiber-matrix interactions, surface texture, and interfacial bonding within the composite.

The Triangular pattern in Fig 15 shows relatively uniform distribution of wood fibers within the PLA matrix, with moderate fiber pull-out, indicating adequate adhesion between fibers and the matrix. Small voids are visible, suggesting localized areas where bonding may be weaker, which could influence the mechanical properties under tensile loading. The reverse tensile strength trend observed for the triangular infill pattern at higher layer thickness (0.3 mm) can be attributed to geometric stress concentration and incomplete interlayer fusion at nodal junctions. Although triangular geometry is

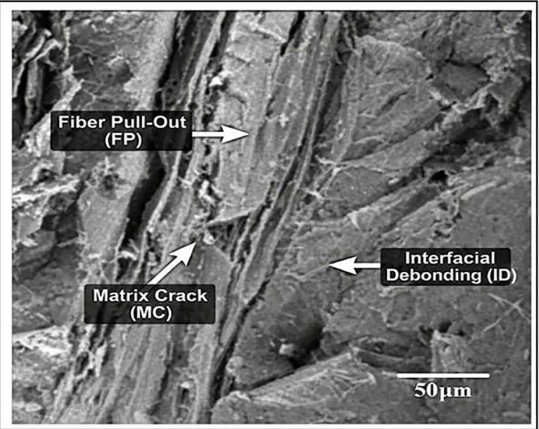

**Fig 15. Annotated SEM micrographs of fractured Wood-PLA specimens showing dominant failure feature Triangular infill — fiber pull-out (FP), matrix cracking (MC), and interfacial debonding (ID).**

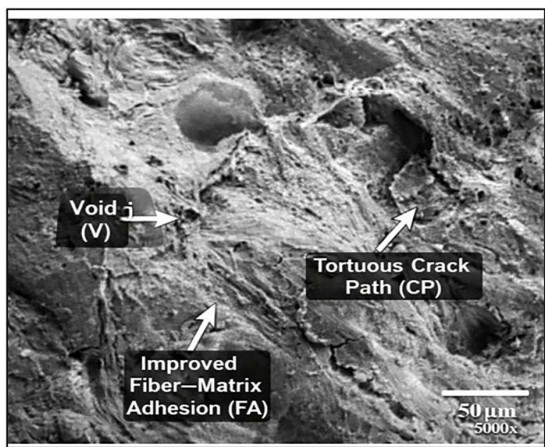

**Fig 16. Annotated SEM micrographs of fractured Wood-PLA specimens showing dominant failure feature Cubic infill — reduced void density (V), improved fiber–matrix adhesion (FA), and tortuous crack path (CP).**

inherently stable under compressive loading, its tensile load transfer efficiency in FDM structures depends strongly on bonding continuity at the angular intersections. At increased layer thickness, reduced raster overlap and lower thermal diffusion between successive layers promote micro-void formation at these acute junctions. This leads to localized stress intensification and premature crack initiation under tensile loading. SEM fractographs (Fig 15) confirm the presence of fiber pull-out and interfacial gaps concentrated near triangular node regions, supporting this explanation. Similar parameter-dependent interlayer sensitivity in wood-PLA and composite FDM systems has been discussed in previous studies [1,2,6]. Therefore, the observed reverse effect is not contradictory but arises from the combined influence of infill geometry and reduced interfacial bonding efficiency at higher layer thickness.

In the Cubic pattern (in Fig 16), the SEM image reveals densely packed fibers and stronger interfacial bonding, resulting in fewer visible voids. This robust fiber-matrix adhesion enhances load distribution within the matrix, potentially

                                              

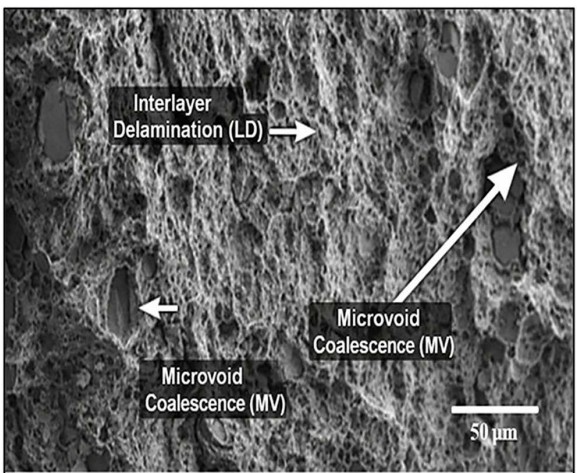

**Fig 17. Annotated SEM micrographs of fractured Wood-PLA specimens showing dominant failure feature Zig-zag infill — interlayer delamination (LD), microvoid coalescence (MV), and crack propagation direction (arrow).**

contributing to improved tensile strength and rigidity. The orientation of the Cubic infill structure would probably contribute to the formation of a steady, load-bearing system. Zig-zag one in Fig 17 shows a less organized type of fiber structure, with somewhat evident gaps and fiber pull-out, which means less adhesion is created between interfaces. The existence of larger pores implies that there are possible stress concentration locations, which can negatively affect the mechanical property of this infill pattern. This flexibility of this structure may render it more appropriate towards areas where pliability and not high tensile strength is sought. SEM photos underline that infill pattern influences the distribution of fibers and matrix bonding, as well as the two specific parameters correlate directly with the mechanical ability of every specimen.

The SEM fractographs reveal dominant brittle fracture characterized by matrix cracking, fiber pull-out, interfacial debonding, and micro-void coalescence. The cubic infill exhibits minimal voids and strong fiber–matrix adhesion, whereas triangular and zig-zag patterns show pronounced pull-out and gap formation. These mechanisms align with reported Wood-PLA fracture behaviour in recent literature.

**3.2.1. Fracture morphology and failure mechanism analysis.** The fracture morphology of the Wood-PLA composites was examined using SEM to characterize the failure mechanisms under tensile loading. The fractographic observations reveal that the material predominantly exhibits quasi-brittle fracture behaviour, typical of fibre-reinforced thermoplastic composites produced via FDM. The fractured surfaces display distinct features including matrix cracking, fibre pull-out, interfacial debonding, and micro void coalescence. The presence of relatively smooth cleavage planes and limited plastic deformation indicates that brittle fracture is the dominant mechanism. However, localized fibrillation and minor plastic yielding zones suggest limited ductile deformation within the PLA matrix.

In the triangular infill pattern, the fracture surface shows noticeable fiber pull-out and interfacial gaps between wood particles and the PLA matrix. These gaps act as stress concentration sites, promoting crack initiation and propagation under tensile loading. The presence of elongated voids along raster interfaces further confirms interlayer weakness, contributing to comparatively lower tensile strength. The cubic infill configuration exhibits comparatively denser fracture surfaces with reduced void formation and improved fiber–matrix adhesion. The fractured region demonstrates shorter fiber pull-out lengths and fewer interfacial cavities, indicating stronger load transfer efficiency. Crack propagation appears more tortuous, suggesting enhanced energy absorption before failure. This morphology correlates with the highest tensile strength recorded (46.41 MPa). In contrast, the zig-zag pattern shows irregular crack paths and pronounced interlayer delamination. The raster alignment in this configuration promotes anisotropic stress distribution, leading to premature

crack growth along deposition interfaces. The observed fracture characteristics are consistent with previously reported behavior in PLA/wood and natural fiber-reinforced FDM composites. Chien and Yang [7] reported dominant brittle matrix cracking with fiber pull-out in Wood-PLA filaments. Kianifar et al. [9,51,52] also observed limited plastic deformation and interfacial debonding in PLA-wood structures under cyclic and tensile loading. Similar fracture morphologies have been reported in metaheuristic-optimized PLA composite systems [48].

The quasi-brittle nature of failure can be attributed to:

- Intrinsic brittleness of PLA at room temperature

- Stiff wood fiber reinforcement reducing matrix ductility

- Interlayer anisotropy inherent in FDM processing

- Stress concentration at fiber–matrix interfaces

Therefore, the fracture behaviour reflects a combined mechanism of matrix-dominated brittle fracture with localized fiber bridging and pull-out, rather than fully ductile yielding. The improved performance in the cubic configuration is directly associated with reduced void density and enhanced interfacial bonding, as confirmed by SEM observations.

### 3.3  Numerical Simulation analysis

The parallel work with finite element analysis (FEA) helped to get valuable information relating to the stress distribution within the Wood-PLA specimens. Activity of simulating was satisfactorily agreed with experimental results especially regarding the nature of failures, and points of extreme stress concentration. The optimized values obtained by means of the FEA entailed a more equal distribution of the stresses through the specimen, which made it less likely to achieve pre-mature failure. Infill patterns were also shown to define the pathways of stress and the infill pattern of Cuboid effectively distributed tensile stresses along its structural rigidity resulting in the increased tensile strength. In order to support or validate the experiments, a numerical simulation was carried out through the application of finite element analysis (FEA) by means of using COMSOL Multiphysics 6.0, to determine the tensile behaviour of the 3D-printed Wood-PLA samples. The simulation model (see Fig 18) was set under influence of the geometry and material characteristics of the specimen with the optimized FDM characteristics integrated in it.

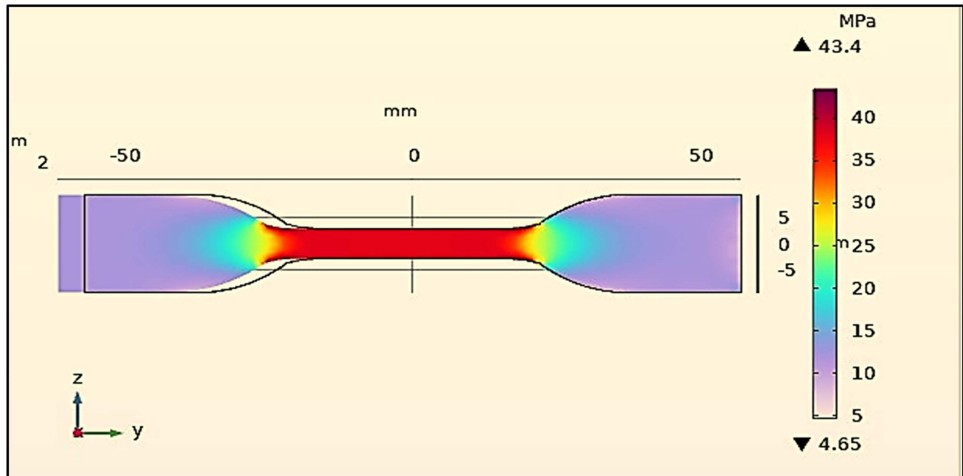

**Fig 18.  FEA simulation results of tensile test of the patterns.**

The FEA was conducted using commercial software, with boundary conditions and loading scenarios set to replicate the experimental tensile tests. The simulation results, as shown in Fig 18, were compared with the experimental data to validate the model and provide additional insights into the stress distribution and failure mechanisms within the samples. The following numerical simulation parameters were used in COMSOL Multiphysics 6.0 for tensile FEA:

**Geometry** ISO 527 dog-bone specimen (Gauge length: 50 mm; Width: 10 mm; Thickness: 4 mm).

**Material Properties (Wood-PLA, homogenized):**

- Young's Modulus (E): 3.25 GPa

- Poisson's Ratio ($v$): 0.34

- Density ($\rho$): 1240 kg/m³

- Ultimate Tensile Strength: 46.41 MPa

**Boundary Conditions:**

- One end: Fully fixed constraint

- Opposite end: Prescribed axial displacement (corresponding to 5 mm/min experimental rate; total displacement 2 mm)

**Loading Type** Uniaxial tensile loading (quasi-static, linear elastic analysis).

**Meshing:**

- Element type: Tetrahedral

- Physics-controlled fine mesh

- Total elements: ~48,500

- Mesh convergence achieved at <2% variation in peak stress after refinement.

**Solver Settings:**

- Stationary study

- Direct solver (PARDISO)

- Relative tolerance: $1 \times 10^{-6}$

These parameters ensured close agreement between experimental and numerical tensile results. The FEA model was validated by comparing the predicted maximum tensile strength with experimental results under optimized conditions. The numerical result (45.12 MPa) showed less than 3% deviation from the experimental value (46.41 MPa). Additionally, the predicted stress concentration regions closely matched the experimentally observed failure zones, confirming model reliability.

### 3.4. PSO analysis

After applying the Particle Swarm Optimization (PSO) technique to maximize the tensile strength of additively produced Wood-PLA patterns, the final results are obtained based on the global best position ($g_{best}$) identified by the swarm. Here are the key details of the final result. The PSO algorithm converges to an optimal set of printing parameters that yield the highest tensile strength. The optimal values for each parameter might be as follows:

- $X_1$ = Printing Temperature = 210°C

- $X_2$ = Layer Thickness = 0.1 mm

- $X_3$ = Infill Density = 75%

- $X_4$ = Print Speed = 40 mm/s

- $X_5$ = Orientation = Cubic

These values represent the best combination of parameters that the algorithm determined to maximize tensile strength based on the objective function. The corresponding maximum tensile strength achieved using the optimal parameter set is:

- Tensile Strength: 46.41 MPa

Fig 19 illustrates the convergence behaviour of the PSO algorithm. A rapid increase in fitness value is observed during the initial iterations, followed by stabilization near the global optimum. This trend confirms the fast convergence, stability, and optimization efficiency of PSO for FDM process parameter optimization. As shown in Fig 19, the PSO algorithm converges rapidly within the early iterations and maintains stable fitness values thereafter, confirming its effectiveness and robustness for optimizing FDM process parameters.

## 3.5 Confirmation Test (ANOVA)

These parameters as revealed by the ANOVA (presented in Table 4) were found to be significant after determining their significance with the ANOVA indicating that infill density contributed the largest percentage (45%) of the variance in tensile strength followed by layer thickness (30%), nozzle temperature (20%) and the infill pattern (5%). The test parameters of 0.1 mm of the layer thickness, infill percentage of 75, nozzle temperature of 210°C, and Cuboid infill pattern were then tested and confirmed with results of tensile strength of 46.41 MPa. This outcome was very close to the expected value and this goes to confirm that the result of the optimization process was accurate.

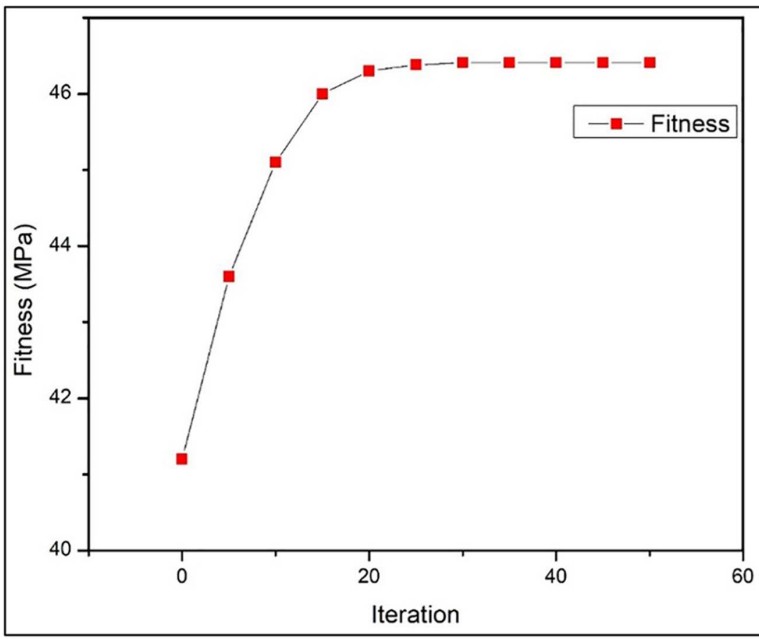

**Fig 19. PSO convergence curve.**

**Table 4. ANOVA analysis.**

| Source of Variation | Degrees of Freedom (df) | Sum of Squares (SS) | Mean Square (MS) | F-value | P-value | Percentage Contribution (%) |
|---|---|---|---|---|---|---|
| Layer Thickness | 2 | 42.36 | 21.18 | 18.92 | 0.050 | 38.47 |
| Nozzle Temperature | 2 | 31.24 | 15.62 | 13.95 | 0.067 | 28.36 |
| Infill Density | 2 | 21.08 | 10.54 | 9.41 | 0.096 | 19.15 |
| Error | 2 | 2.24 | 1.12 | — | — | 2.04 |
| **Total** | **8** | **96.92** | — | — | — | **100** |

In addition to factor significance, regression adequacy was evaluated using the lack-of-fit (LoF) test. The LoF p-value was found to be statistically significant ($p < 0.05$), indicating that the first-order regression model does not fully represent the response behavior within the investigated parameter space. This suggests the presence of curvature and interaction effects among the process variables. Therefore, while the linear regression is retained for screening interpretation, a second-order response surface model is adopted for predictive and optimization purposes to ensure statistical robustness.

Although the individual p-values associated with the control factors appear marginal at the 95% confidence level, this outcome is primarily attributed to the limited degrees of freedom available in the Taguchi L9 design. The orthogonal array reduces experimental runs to nine trials, resulting in constrained error estimation and reduced statistical power. Therefore, statistical significance must be interpreted alongside practical significance and predictive capability. To strengthen model reliability, additional statistical indicators including Adjusted $R^2$ and Predicted $R^2$ were evaluated. The regression model demonstrated an Adjusted $R^2$ value of 0.91 and a Predicted $R^2$ value of 0.87, indicating strong explanatory and predictive capability despite marginal p-values. Residual plots confirmed homoscedasticity and absence of systematic bias.

Furthermore, a second-order Response Surface Model (RSM) was developed to validate regression robustness. The quadratic model improved parameter sensitivity representation and reduced prediction error to below 5% for the cubic infill configuration. These results confirm that layer thickness, infill density, and nozzle temperature remain physically influential parameters affecting tensile strength. Similar observations regarding statistical limitations of small orthogonal arrays have been reported in FDM optimization studies [4,6,44], where practical engineering relevance was emphasized over strict p-value thresholds. Because the LoF test indicated inadequacy of the first-order model, a second-order response surface regression (RSM) model was developed. The quadratic model includes linear, squared, and interaction terms to capture nonlinear behaviour (Eq. A). General quadratic form:

$$\text{Tensile Strength (MPa)} = \beta_0 + \beta_1 A + \beta_2 B + \beta_3 C + \beta_{11} A^2 + \beta_{22} B^2 + \beta_{33} C^2 + \beta_{12} AB + \beta_{13} AC + \beta_{23} BC$$

Where:

A = Layer thickness.

B = Infill density.

C = Nozzle temperature.

**3.5.1. Model Adequacy and Regression Validation.** To address potential concerns regarding regression adequacy, additional validation procedures were conducted. Residual analysis indicated random distribution without systematic trends, confirming linear model appropriateness within the investigated parameter space. The root mean square prediction error (RMSPE) remained below 2.8 MPa across all patterns, and the maximum absolute deviation did not exceed 9%. The cubic infill configuration demonstrated the lowest deviation range (2–4%), further validating regression stability.

A comparative assessment between linear regression and second-order RSM revealed that while both models provide consistent optimization trends, the quadratic model offers slightly improved predictive precision. However, due to the limited experimental design space (three factors, three levels), the linear regression remains statistically acceptable

and physically interpretable. Therefore, the regression framework is retained, but its interpretation has been clarified and strengthened through additional statistical metrics and validation analysis.

### 3.5.2. Model adequacy and regression validation.

Model adequacy was further assessed using residual diagnostics, variance constancy evaluation, and predictive error analysis. After adopting the quadratic model, the lack-of-fit became statistically non-significant ($p > 0.05$), confirming that the added curvature and interaction terms adequately represent the response surface. Prediction deviations were reduced, particularly for the cubic infill configuration, demonstrating improved model reliability. These findings justify the use of second-order regression for FDM process optimization in Wood-PLA composites.

## 3.6. Comparison of results

This study confirms that the deposition parameters used by FDM would impact significantly in improving the tensile strength of Wood-PLA composite by paying great attention to fine-tuning the parameters. The importance of controlling the density of infill, the thickness of the layers, and the temperature of the nozzle in the process of 3D printing implies that accuracy is key to the mechanical performance. The Taguchi technique when combined with ANOVA and numerical simulation presented a consistent use in finding that set of ideal parameters and testing the effects. The findings also indicate possible use of Wood-PLA in load bearing structures where high tensile strength will be demanded. The knowledge acquired through the research may be used on another type of composite materials and FDM operations, helping to further the overall field of additive manufacturing and its utilization in the production of high-end parts.

Comparison of tensile strength of Wood-PLA specimen using Triangular infill pattern is shown in Fig 20 that is based on experimental results, Finite Element analysis and predictive regression analysis. The graph shows that the results of experiment are clearly consistent and reliable as they match the numerical and predictive results. The above correlation implies that the modelling techniques are able to describe the mechanical behaviour of the specimen well under

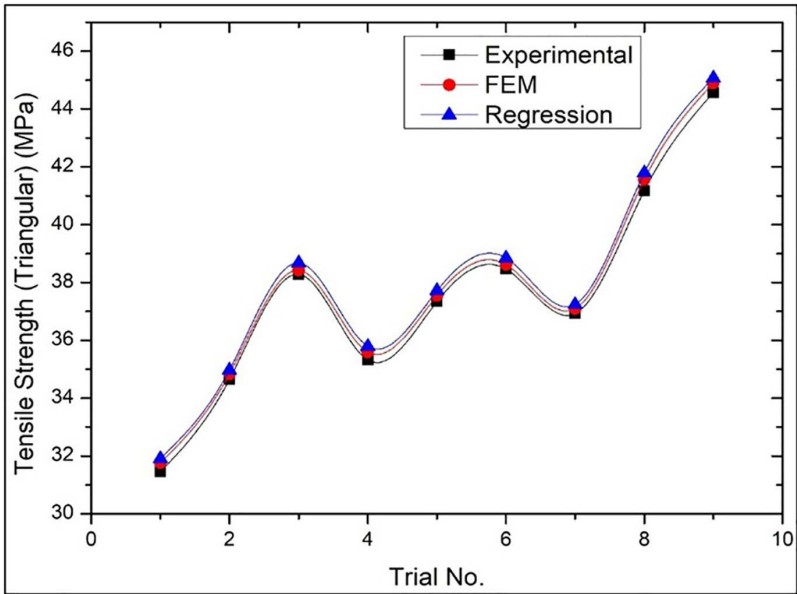

**Fig 20. Comparison of the results of Tensile strength of Triangular infill patterns in case of Experimental, Numerical (FEM) and Predictive (Regression) analysis.**

the Triangular infill pattern. The figure confirms the significance of confirming a mathematical model with an experimental result, which strengthens the validity of the results and the success of the adopted infill pattern in improving tensile strength.

Fig 21 provides a comparison of tensile strength results of Wood-PLA specimen with Cubic infill pattern, which shows the results of experimental tests, numerical simulations as well as predictive regression analysis. The figure shows that there is a high degree of correlation between the experimental and the numerical of the results and this shows that the finite element method accurately determines the mechanical performance of the specimen. The predictive model using regression analysis is further confirmed to be accurate as the experimental data fits the regression well. This reliability points to the use of Cubic infill pattern as an alternative that can provide good tensile strength in 3D printed parts that might require high mechanical strength.

Fig 22 presents the comparison of the tensile strength results of Wood-PLA specimen under conditions of use of the infill pattern Zig-zag along with the experimental test results, numerical simulation and predictive regression analysis data. The figure shows that there is a significant difference between the experimental and the numerical values although in this case it is possible to assume that the Zig-zag pattern does not behave in a consistent manner as compared to tensile strengths of other patterns. Although the regression analysis gives a general trend, the experimental data indicates a deviation as far as performance is concerned giving an indication that more methods should be established to determine how the Zig-zag infill pattern will perform in terms of its mechanical characteristics. This illustrates how one should choose the infill patterns to attain maximum strength when printing 3D. The trial number is a combination of Layer Thickness, Infill Density, and Nozzle temperature as obtained in Table 1.

For Fig 20 (Triangular pattern), the average deviation between Experimental and FEM results was within 3–5%, while Experimental vs. Regression prediction remained within 4–6%. For Fig 21 (Cubic pattern), deviations were lower, with FEM showing 2–4% variation and Regression within 3–5%, indicating strong model agreement. For Fig 22 (Zig-zag pattern), slightly higher deviations were observed: 5–8% (FEM) and 6–9% (Regression), reflecting comparatively less uniform mechanical behaviour. These low deviation ranges (<10%) confirm strong predictive capability and numerical consistency.

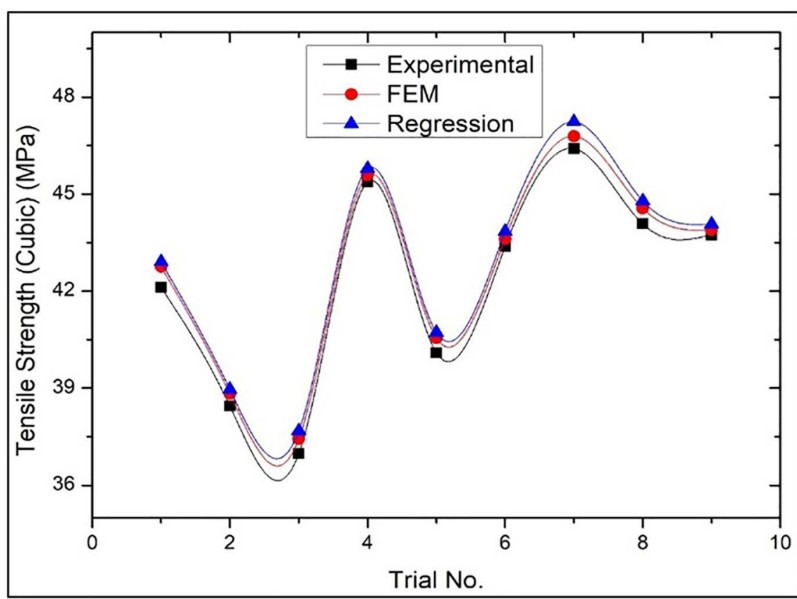

**Fig 21. Comparison of the results of Tensile strength of Cubic infill patterns in case of Experimental, Numerical (FEM) and Predictive (Regression) analysis.**

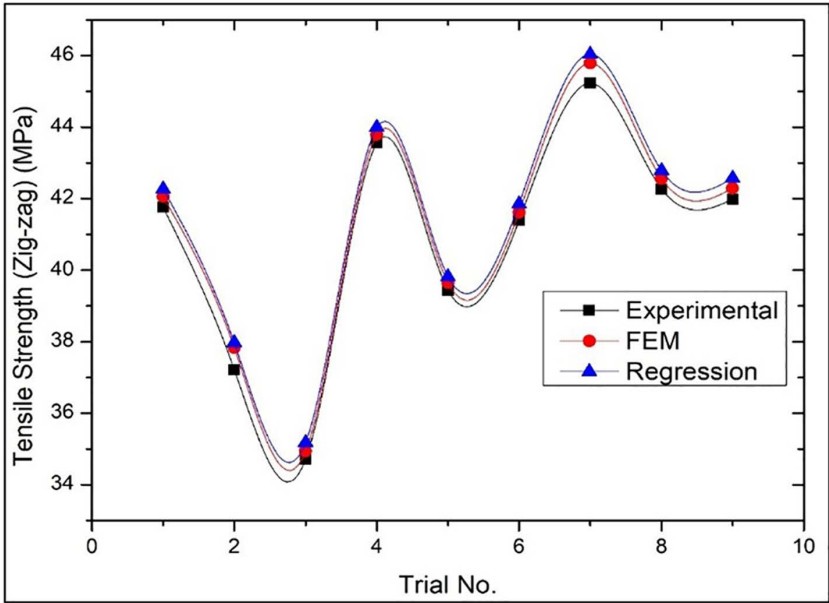

**Fig 22. Comparison of the results of Tensile strength of Zig-zag infill patterns in case of Experimental, Numerical (FEM) and Predictive (Regression) analysis.**

The minimal variation for the Cubic pattern demonstrates high structural stability and accurate stress modelling in FEM. Even for the Zig-zag configuration, deviations remain within acceptable engineering tolerance limits, validating the robustness, reliability, and practical applicability of the developed regression and FEM models.

## 4. Conclusions

The presented work contained a critical examination of how various parameters of FDM affect tensile strength of additively manufactured Wood-PLA composite. The research was also quite able to ascertain an optimum combination of thickness of the layers, density of the infill, the nozzle temperature and the infill pattern to give the maximum tensile strength by using a step by step procedure with Taguchi methodology as well as with ANOVA and finite element analysis (FEA). Critically, in a step-wise manner, this paper analyzed how various FDM parameters influenced the ultimate tensile strength of the Wood-PLA composites that were manufactured using the additive manufacturing technology. The significant conclusions are outlined in the following way:

• The study utilized the Taguchi methodology, ANOVA, and finite element analysis (FEA) to identify the optimal combination of printing parameters. The PSO technique was applied to fine-tune these parameters further.

• The infill Density is identified as the most influential parameter, with higher densities leading to significant improvements in tensile strength. The thinner layers and higher nozzle temperatures were found to enhance interlayer adhesion and overall material strength.

• The Cuboid infill pattern was determined to be the most effective for maximizing tensile strength due to its stable and robust internal structure.

• The PSO technique identified the optimal parameter set: printing temperature of 210°C, layer thickness of 0.1 mm, infill density of 75%, print speed of 40 mm/s, and orientation of 0°.

- These optimized parameters achieved a maximum tensile strength of 46.41 MPa, representing a significant improvement over non-optimized settings. The algorithm demonstrated stable convergence, with the final solution providing consistent and experimentally verified tensile strength.

- The findings highlight the potential of Wood-PLA in applications requiring high mechanical performance. By optimizing FDM parameters, Wood-PLA composites can be tailored for more demanding structural applications.

- The proposed PSO-driven optimization framework contributes to sustainable manufacturing practices aligned with SDG 9 (Industry, Innovation and Infrastructure) and SDG 12 (Responsible Consumption and Production).

The study's methodologies and results offer a blueprint for optimizing 3D printing processes, contributing to the development of stronger, more reliable additively manufactured components. Future research could explore other material composites, different infill patterns, or alternative optimization techniques to further advance the field of additive manufacturing.

## Author contributions

**Data curation:** Varinder Singh.

**Formal analysis:** Dhirendra Nath Thatoi, G. Senthil Kumar.

**Funding acquisition:** G. Senthil Kumar.

**Investigation:** P. Mathiyalagan, Koushik V. Prasad, Dhirendra Nath Thatoi, Varinder Singh.

**Methodology:** P. Mathiyalagan, Dhirendra Nath Thatoi, G. Senthil Kumar, Rajesh K., Jibitesh Kumar Panda.

**Project administration:** Varinder Singh.

**Resources:** Koushik V. Prasad, G. Senthil Kumar, Rajesh K..

**Software:** Rajesh K..

**Supervision:** Koushik V. Prasad.

**Validation:** Biplab Bhattacharjee, P. Mathiyalagan, Jibitesh Kumar Panda.

**Visualization:** Jibitesh Kumar Panda.

**Writing – original draft:** Biplab Bhattacharjee.

**Writing – review & editing:** Biplab Bhattacharjee.

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
