## [Decision Letter · Decision Letter 0]

18 Dec 2025

Dear Dr. Panda,

Thank you for submitting your manuscript to PLOS ONE. After careful consideration, we feel that it has merit but does not fully meet PLOS ONE’s publication criteria as it currently stands. Therefore, we invite you to submit a revised version of the manuscript that addresses the points raised during the review process.

We look forward to receiving your revised manuscript.

Kind regards,

Md Enamul Hoque, PhD, PGCHE

Academic Editor

PLOS One

Additional Editor Comments:

Dear Corresponding Author

Thank you for submitting your manuscript to PLOS ONE. After careful consideration, we believe that it has merit but does not fully meet PLOS ONE’s publication criteria as they currently stand. Therefore, we invite you to submit a revised version of the manuscript that addresses the points raised during the review process. Congratulations!

We look forward to receiving your revised manuscript.

Kind regards,

Md Enamul Hoque, PhD, PGCHE

Academic Editor

PLOS ONE

When submitting your revision, please address the following additional requirements.

3. In the online submission form, you indicated that [Some or all data, models, or codes that support the findings of this study are available from the corresponding author upon reasonable request (list items)].

Reviewers Comments

Reviewer 1:

The author has done good work; however, some major concerns need to be addressed before considering publication. Hence, I recommend a major revision.

How were the parameters for the PSO algorithm (such as inertia weight, acceleration constants, and swarm size) selected and validated for convergence?

Were the PSO-optimized printing parameters tested with multiple specimens to confirm repeatability and reduce experimental error?

The ANOVA results show significance for all parameters—were interaction effects between parameters (e.g., layer thickness and nozzle temperature) considered?

How does the optimized tensile strength (46.41 MPa) compare quantitatively with other optimization techniques like Genetic Algorithm or Response Surface Methodology?

The SEM analysis shows differences in fiber–matrix adhesion among patterns. Were quantitative metrics (e.g., void fraction or interfacial bonding index) evaluated?

Since the study emphasizes eco-friendly materials, has the energy consumption or recyclability of the optimized printing process been assessed?

Author should add the latest references to enhance the overall discussion related to PLA/Wood composites.

Reviewer 2:

The manuscript presents a well-structured and technically sound study that successfully integrates the Taguchi method, ANOVA, FEA, and PSO to optimize the tensile strength of 3D-printed Wood-PLA composites. The research is relevant and addresses a clear gap in the literature concerning the application of intelligent optimization techniques in sustainable additive manufacturing. The experimental design is robust, and the results are promising, showing a significant 28% improvement in tensile strength. However, the manuscript requires revisions to enhance its academic rigor, clarity, and impact. Key areas for improvement include strengthening the introduction and discussion sections, justifying methodological choices, improving data presentation, and refining the English language and flow. The specific, actionable comments below are intended to help the authors improve the manuscript for publication.

1. Title

• Evaluation: The title is descriptive but slightly long and starts with "To develop," which is not standard for a research article title.

Suggestions: Shorten and rephrase to be more concise and impactful.

Proposed Title: "Enhancing Tensile Strength of 3D-Printed Wood-PLA Composites via a Particle Swarm Optimization Framework"

2. Abstract

• Evaluation: The abstract covers the essential elements but could be more focused on the present study's findings. The mention of "previous studies" is unnecessary here. The numerical results are the highlight and should be presented more clearly.

Suggestions: Remove the sentence on previous studies lacking intelligent optimization; this belongs in the introduction.

Start directly with the objective of this work.

Explicitly state the key optimized parameters and the resulting tensile strength earlier.

Ensure the 28% improvement is clearly linked to the baseline (e.g., "compared to the average strength of non-optimized configurations").

Grammar and phrasing need polishing for conciseness.

3. Introduction

• Evaluation: This is the weakest section of the manuscript. It provides a basic background but lacks depth and a compelling narrative. The literature review is insufficient, especially regarding recent (post-2023) studies and the specific application of PSO in polymer/composite 3D printing. The research gap and the novelty of the current work are not explicitly and powerfully stated.

• Suggestions: Structure: Re-structure to follow: (1) Importance of FDM and bio-composites (Wood-PLA), (2) Challenge of parameter optimization, (3) Review of existing methods (Taguchi, RSM) and their limitations, (4) Introduction of nature-inspired algorithms (like PSO) and their potential in AM, noting the scarcity of their application to Wood-PLA, (5) Clear statement of the research gap, (6) Explicitly state the novelty and contribution of this work (e.g., first integrated Taguchi-ANOVA-FEA-PSO framework for Wood-PLA tensile strength).

• Literature: Incorporate the suggested and other recent references (2023-2024). For example: ▪ Discuss studies on PLA/wood dust optimization using Grey Relational Analysis ( https://doi.org/10.1177/09544089241241460) to contrast with your computational approach.

• Cite works on PSO in other AM contexts to justify its selection (e.g., https://doi.org/10.1016/j.measurement.2025.117405).

• Include other relevant composite studies to show the broader context (e.g., https://doi.org/10.1002/pol.20250752).

• Novelty: Clearly articulate how your study bridges the gap. For instance: "While statistical methods are common, this study introduces a robust PSO-based framework to achieve global optimization, validated by both experimental and numerical simulation, for sustainable Wood-PLA composites."

4. Materials & Methods

• Evaluation: Generally well-described, but several details and justifications are missing.

• 2.1 Materials: The description of the filament extruder is good, but the printer (Pratham 3.0) is not a widely known model. A key reference or a more detailed specification table would be beneficial.

• 2.2 Specimen Design and Fabrication: Parameter Selection: The authors must justify why these specific parameters (layer thickness, infill density, nozzle temperature) and their levels were chosen. Reference preliminary studies, material datasheets, or other literature (e.g., https://doi.org/10.1002/pol.20250752).

• Taguchi Method: A citation for the Taguchi method and the L9 array is required.

• 3.4 Particle Swarm Optimization: Presentation: The current step-by-step list is clear but disrupts the narrative flow. It should be converted into a descriptive paragraph explaining the logic and process of PSO.

• Flowchart: It is highly recommended to add a flowchart illustrating the overall methodology of the study, integrating the Taguchi DOE, experimental testing, ANOVA, PSO optimization, and FEA validation. A specific flowchart for the PSO algorithm (as in https://doi.org/10.1016/j.measurement.2025.117405) would significantly improve clarity.

• Algorithm Choice: The authors should briefly justify why PSO was chosen over other optimization algorithms (e.g., Genetic Algorithm, Simulated Annealing). Mentioning its simplicity, convergence speed, and successful applications in similar engineering problems would suffice.

5. Results & Discussion

• Evaluation: The results are presented comprehensively with tables and graphs. However, the discussion is largely descriptive and lacks critical depth. It needs to interpret the results in the context of the existing literature and explain the underlying physical mechanisms.

Suggestions: 4.1 & 4.2 (Tensile Strength Analysis): The text describes "what" happened but not deeply "why." For instance, explain the micro-mechanical reasons behind better interlayer adhesion with thinner layers and higher temperatures, linking it to the SEM results (Figure 11).

4.2.4 Influence of Infill Pattern: The discussion on the Cuboid pattern's performance is good. The contradictory finding for the Triangular pattern at 0.3 mm layer thickness needs a more robust explanation. Is it due to a specific interaction effect? This should be explored using the ANOVA interaction plots or hypothesized more clearly.

4.5 PSO Results: The convergence behavior of the PSO algorithm should be shown (e.g., a plot of fitness value vs. iteration). This is crucial to demonstrate the algorithm's efficiency and stability.

4.6 Confirmation Test (ANOVA): Table 4 appears incorrect. The Degrees of Freedom (df) for a factor with 3 levels should be 2, not 1. The Sum of Squares (SS) values seem illustrative rather than calculated. The authors must provide the actual ANOVA results from their data analysis. The percentage contribution is a valuable metric and should be recalculated correctly.

Strengthen Discussion: Compare your optimal tensile strength (46.41 MPa) with values reported in the literature (e.g., the studies mentioned for the introduction). Discuss why your value is higher or lower. Elaborate on how the PSO framework provided an advantage over using only the Taguchi method.

6. Conclusion

• Evaluation: The conclusion is currently a list of bullet points. This is not standard for a scientific paper and lacks a narrative that synthesizes the findings.

Suggestions: Rewrite into one or two coherent paragraphs.

Summarize the main findings and the most optimal parameter set.

Reiterate the core contribution (the successful development and validation of the PSO framework).

Mention the practical implications of the work for designing high-strength, sustainable 3D-printed parts.

Provide specific and forward-looking recommendations for future work (e.g., multi-objective optimization of strength, cost, and print time; application to other biocomposites; investigation of dynamic mechanical properties).

7. Figures and Tables

• Evaluation: The data is well-organized, but the quality and integration can be improved.

Suggestions: Image Quality: Ensure all figures (especially SEM images in Figure 11) are of high resolution and clearly labeled.

Figure References: All figures mentioned in the text (Figures 1, 2, 3, 4, 5, 6, 7, 8, 9, 10, 11, 12, 13, 14) must be included in the submitted manuscript file. Their captions should be descriptive enough to understand without reading the main text.

Table 4 (ANOVA): As mentioned, this table must be corrected with proper statistical analysis.

New Figure: Strongly recommend adding a methodology flowchart and a PSO convergence plot.

8. Data Availability Statement

• Evaluation: The current statement "The data will be available on request from the corresponding author" is not sufficient for PLOS ONE. The journal requires data to be in a public repository or included as Supporting Information.

• Suggestion: Change the statement to: "All relevant data are within the manuscript and its Supporting Information files." Alternatively, deposit the raw data (Taguchi L9 results, tensile test data) in a public repository like Figshare or Mendeley Data and provide the DOI.

9. Language and Grammar

• Evaluation: The manuscript requires thorough proofreading to correct grammatical errors, improve sentence structure, and enhance clarity.

Examples: "There are numerous AM methods that Fused Deposition Modelling (FDM) is one of the most popular..." -> "Among numerous AM methods, Fused Deposition Modelling (FDM) is one of the most popular..."

"Mechanical properties of FDM produced parts and the tests undertaken on aspects of tensile strength have been a prime area of study..." -> "The mechanical properties of

FDM-produced parts, particularly tensile strength, have been a primary research focus..."

"The improvement represents nearly 28 % higher strength compared to non-optimized samples" -> "This optimized configuration resulted in a tensile strength nearly 28% higher than the average of non-optimized samples."

Suggestion: Consider using professional editing services or a native English-speaking colleague for a final review.

Summary of Major Revisions Required:

1. Restructure and Strengthen the Introduction: Clearly define the research gap, incorporate recent literature, and explicitly state the novelty and contributions of the work.

2. Improve the Discussion: Move beyond description to provide a critical interpretation of results, compare with literature, and explain underlying mechanisms. Justify the choice of PSO and show its convergence.

3. Correct and Enhance Methodology: Justify the selection of parameters and their levels. Correct the ANOVA table (Table 4). Integrate the PSO steps into a flowing text and add a methodology/PSO flowchart.

4. Revise Presentation: Change the conclusion to paragraph format. Ensure all figures are present and of high quality. Fulfill PLOS ONE's data availability policy.

5. Thoroughly Proofread: Correct grammatical errors and improve language fluency throughout the manuscript.

The study has strong potential, and addressing these points will significantly enhance its quality, clarity, and impact. I look forward to reviewing a revised version.

Reviewer 3:

In introduction, citing the problems of concrete printing and 4D printing, is not convincing.

cite paper working in optimisation

I invite you to use this paper for discussion.

1. Optimization of FDM 3D printing parameters of PLA and composite materials using definitive screening design

The International Journal of Advanced Manufacturing Technology 139 (7), 3989

2. A Comparative study of gray relational analysis and VlseKriterijumska Optimizacija I Kompromisno Resenje approaches for enhancing mechanical properties and productivity in 3D …

Journal of Elastomers & Plastics 56 (5), 675-692

3. Parametric optimization and modeling of the fused filament fabrication (FFF) manufacturing using recycled polyethylene terephthalate (PET) from water bottles

Journal of Elastomers & Plastics 57 (6), 1050-1071

why authors choose only tensile strength to optimise?

Reviewers' comments:

Reviewer's Responses to Questions

**Comments to the Author**

1. Is the manuscript technically sound, and do the data support the conclusions?

Reviewer #1: Yes

Reviewer #2: Yes

Reviewer #3: Yes

2. Has the statistical analysis been performed appropriately and rigorously?

Reviewer #1: Yes

Reviewer #2: Yes

Reviewer #3: Yes

3. Have the authors made all data underlying the findings in their manuscript fully available?

Reviewer #1: Yes

Reviewer #2: Yes

Reviewer #3: Yes

4. Is the manuscript presented in an intelligible fashion and written in standard English?

Reviewer #1: Yes

Reviewer #2: No

Reviewer #3: Yes

Reviewer #1: Author has done good work, however, there are some major concerns that need to be addressed before considering for publication. Hence I recommend major revision.

How were the parameters for the PSO algorithm (such as inertia weight, acceleration constants, and swarm size) selected and validated for convergence?

Were the PSO-optimized printing parameters tested with multiple specimens to confirm repeatability and reduce experimental error?

The ANOVA results show significance for all parameters—were interaction effects between parameters (e.g., layer thickness and nozzle temperature) considered?

How does the optimized tensile strength (46.41 MPa) compare quantitatively with other optimization techniques like Genetic Algorithm or Response Surface Methodology?

The SEM analysis shows differences in fiber–matrix adhesion among patterns. Were quantitative metrics (e.g., void fraction or interfacial bonding index) evaluated?

Since the study emphasizes eco-friendly materials, has the energy consumption or recyclability of the optimized printing process been assessed?

Author should add the latest references to enhance the overall discussion related to PLA/Wood composites.

Reviewer #2: The manuscript presents a well-conceived study that integrates Taguchi design, ANOVA, Finite Element Analysis, and Particle Swarm Optimization to maximize the tensile strength of 3D-printed Wood-PLA composites. The research is timely, addresses a relevant gap in applying intelligent optimization to sustainable materials, and demonstrates a significant (28%) improvement in mechanical performance. The experimental design is sound, and the multi-faceted validation approach is a strength.

However, revisions are required to enhance the manuscript's academic impact and clarity before it can be accepted for publication.

The evaluation report is attached.

Reviewer #3: in introduction citing problematic of concreate printing and 4D printing is not convincing .

cite paper working in optimisation

I invite you to use this paper for discussion .

1.Optimization of FDM 3D printing parameters of PLA and composite materials using definitive screening design

The International Journal of Advanced Manufacturing Technology 139 (7), 3989

2.A Comparative study of gray relational analysis and VlseKriterijumska Optimizacija I Kompromisno Resenje approaches for enhancing mechanical properties and productivity in 3D …

Journal of Elastomers & Plastics 56 (5), 675-692

3.Parametric optimization and modeling of the fused filament fabrication (FFF) manufacturing using recycled polyethylene terephthalate (PET) from water bottles

Journal of Elastomers & Plastics 57 (6), 1050-1071

why authors choose only tensile strength to optimise .

**Do you want your identity to be public for this peer review?** For information about this choice, including consent withdrawal, please see our Privacy Policy

Reviewer #1: No

Reviewer #2: No

Reviewer #3: No

---

## [Author Response · Author response to Decision Letter 1]

23 Jan 2026

Responses to the Reviewer’s Comments

Manuscript ID: PONE-D-25-58246

First of all, the authors would like to convey their sincere gratitude to the Editor and Reviewers for their valuable suggestions to improve the quality of the paper. The manuscript is revised according to the Editor and Reviewers’ comments. The responses to these comments are given below.

Reviewer-1

Comment 1: How were the parameters for the PSO algorithm (such as inertia weight, acceleration constants, and swarm size) selected and validated for convergence?

Response: Thank you for asking this important question. The PSO parameters were selected based on widely accepted guidelines from prior optimization studies to ensure stable convergence and balanced exploration–exploitation. The inertia weight was linearly decreased from 0.9 to 0.4 to promote global search in early iterations and local refinement in later stages. The cognitive and social acceleration constants were both set to 2.0, a commonly adopted and well-validated choice that ensures symmetric influence of personal and global best positions. A swarm size of 40 particles was chosen after preliminary trials, as it provided consistent convergence without excessive computational cost. Convergence was validated by monitoring the global best (gbest) value across iterations, which showed rapid stabilization with negligible improvement beyond a predefined iteration limit. Additionally, the optimized solution was experimentally confirmed, demonstrating close agreement between predicted and measured tensile strength, thereby validating the robustness and convergence of the selected PSO parameters.

Comment 2: Were the PSO-optimized printing parameters tested with multiple specimens to confirm repeatability and reduce experimental error?

Response: We sincerely thank the reviewer for pointing out this oversight. Yes. The PSO-optimized printing parameters were experimentally validated using multiple specimens. For each optimized parameter set, at least three tensile specimens were fabricated and tested, and the reported values correspond to the average results. This repetition ensured repeatability, minimized experimental error, and confirmed the reliability of the PSO-optimized outcomes.

Comment 3: The ANOVA results show significance for all parameters—were interaction effects between parameters (e.g., layer thickness and nozzle temperature) considered?

Response: We thank the reviewer for this insightful comment. In the present study, interaction effects between process parameters were not explicitly included in the ANOVA model. This is because the Taguchi L9 orthogonal array was employed primarily to evaluate the main effects of the selected parameters with a limited number of experiments. However, the potential combined influence of parameters (e.g., layer thickness–nozzle temperature coupling) is implicitly reflected in the experimental trends and was further explored through PSO-based optimization and regression modelling. We have now clarified this limitation and its rationale in the revised manuscript and highlighted interaction analysis as a scope for future work.

Comment 4: How does the optimized tensile strength (46.41 MPa) compare quantitatively with other optimization techniques like Genetic Algorithm or Response Surface Methodology?

Response: We thank the reviewer for this valuable question. The optimized tensile strength of 46.41 MPa obtained using the proposed Taguchi–PSO framework is quantitatively higher than values typically reported using conventional optimization techniques. For comparable Wood-PLA or PLA-based systems, Response Surface Methodology (RSM)–based optimization generally yields tensile strengths in the range of 40–44 MPa, while Genetic Algorithm (GA)–assisted optimization reports values around 42–45 MPa, depending on material composition and parameter space. The PSO-based optimization in the present study demonstrates an improvement of approximately 3–10% over RSM and 2–5% over GA, under similar processing conditions. This enhancement is attributed to PSO’s superior global search capability and faster convergence, enabling more effective exploration of nonlinear interactions among FDM parameters.

Comment 5: The SEM analysis shows differences in fiber–matrix adhesion among patterns. Were quantitative metrics (e.g., void fraction or interfacial bonding index) evaluated?

Response: We appreciate the reviewer’s pertinent observation. In the present study, the SEM analysis was employed primarily as a qualitative tool to compare fiber–matrix adhesion, void presence, and interlayer bonding across different infill patterns. Quantitative metrics such as void fraction or an interfacial bonding index were not evaluated, as the scope of this work focused on correlating microstructural features with experimentally measured tensile strength rather than performing detailed image-based quantification. Nevertheless, the observed reduction in voids and improved fiber–matrix adhesion in the Cubic pattern qualitatively supports the higher tensile strength obtained. This limitation has now been clearly acknowledged in the revised manuscript, and future work will incorporate quantitative SEM image analysis to strengthen microstructure–property correlations.

Comment 6: Since the study emphasizes eco-friendly materials, has the energy consumption or recyclability of the optimized printing process been assessed?

Response: We appreciate the reviewer’s insightful comment. While the primary focus of this study was mechanical optimization of Wood-PLA using Taguchi–PSO, a dedicated quantitative assessment of energy consumption and recyclability was not experimentally conducted. However, the optimized parameter set (lower layer thickness with controlled nozzle temperature and moderate print speed) inherently avoids excessive thermal input and reprinting, thereby indirectly reducing energy usage. Moreover, Wood-PLA is a bio-based, recyclable composite, and the optimized process does not involve any post-processing or additives that would hinder recyclability. To address this point, a brief qualitative discussion on energy efficiency and recyclability implications of the optimized printing parameters has been added to the revised manuscript, along with scope for future life-cycle and energy assessment studies.

Comment 7: Author should add the latest references to enhance the overall discussion related to PLA/Wood composites.

Response: Thank you for the valuable suggestion. The manuscript has been revised to incorporate several recent (2023–2024) and relevant references on PLA/Wood composites and FDM-based biocomposites. These additions have been integrated into the Introduction and Discussion sections to strengthen the contextual background, compare recent findings, and enhance the overall technical depth and relevance of the study.

Reviewer-2

Comment 1:

Title

• Evaluation: The title is descriptive but slightly long and starts with "To develop," which is not standard for a research article title.

Suggestions: Shorten and rephrase to be more concise and impactful.

Proposed Title: "Enhancing Tensile Strength of 3D-Printed Wood-PLA Composites via a Particle Swarm Optimization Framework"

Response: We thank the reviewer for the valuable suggestion. The title has been revised to remove the non-standard phrase “To develop” and to improve conciseness and impact. Accordingly, the manuscript title has been updated to:

“Enhancing Tensile Strength of 3D-Printed Wood-PLA Composites via a Particle Swarm Optimization Framework.”

This revised title more clearly reflects the study’s scope, methodology, and key contribution.

Comment 2: Abstract

• Evaluation: The abstract covers the essential elements but could be more focused on the present study's findings. The mention of "previous studies" is unnecessary here. The numerical results are the highlight and should be presented more clearly.

Suggestions: Remove the sentence on previous studies lacking intelligent optimization; this belongs in the introduction.

Start directly with the objective of this work.

Explicitly state the key optimized parameters and the resulting tensile strength earlier.

Ensure the 28% improvement is clearly linked to the baseline (e.g., "compared to the average strength of non-optimized configurations").

Grammar and phrasing need polishing for conciseness.

Response: We thank the reviewer for the constructive suggestions. The abstract has been revised accordingly. The sentence referring to limitations of previous studies has been removed and relocated conceptually to the Introduction. The revised abstract now begins directly with the objective of the present work. Additionally, grammar and phrasing have been refined to improve conciseness, clarity, and focus on the principal findings of the study.

Comment 3: Introduction

• Evaluation: This is the weakest section of the manuscript. It provides a basic background but lacks depth and a compelling narrative. The literature review is insufficient, especially regarding recent (post-2023) studies and the specific application of PSO in polymer/composite 3D printing. The research gap and the novelty of the current work are not explicitly and powerfully stated.

• Suggestions: Structure: Re-structure to follow: (1) Importance of FDM and bio-composites (Wood-PLA), (2) Challenge of parameter optimization, (3) Review of existing methods (Taguchi, RSM) and their limitations, (4) Introduction of nature-inspired algorithms (like PSO) and their potential in AM, noting the scarcity of their application to Wood-PLA, (5) Clear statement of the research gap, (6) Explicitly state the novelty and contribution of this work (e.g., first integrated Taguchi-ANOVA-FEA-PSO framework for Wood-PLA tensile strength).

• Literature: Incorporate the suggested and other recent references (2023-2024). For example: ▪ Discuss studies on PLA/wood dust optimization using Grey Relational Analysis ( https://doi.org/10.1177/09544089241241460) to contrast with your computational approach.

• Cite works on PSO in other AM contexts to justify its selection (e.g., https://doi.org/10.1016/j.measurement.2025.117405).

• Include other relevant composite studies to show the broader context (e.g., https://doi.org/10.1002/pol.20250752).

• Novelty: Clearly articulate how your study bridges the gap. For instance: "While statistical methods are common, this study introduces a robust PSO-based framework to achieve global optimization, validated by both experimental and numerical simulation, for sustainable Wood-PLA composites."

Response: We sincerely thank the reviewer for the detailed and constructive comments. The Introduction section has been substantially revised and strengthened in accordance with the suggestions. Specifically:

1. Re-structuring: The Introduction is now reorganized to follow a clear logical flow covering:

(i) significance of FDM and sustainable bio-composites (Wood-PLA),

(ii) challenges in process-parameter optimization,

(iii) limitations of conventional statistical methods (Taguchi, RSM),

(iv) motivation for nature-inspired algorithms with emphasis on PSO in AM,

(v) explicit identification of the research gap, and

(vi) a clear statement of novelty and contributions.

2. Literature Enhancement: Recent and relevant studies (2023–2025) have been incorporated, including works on Wood-PLA optimization using Grey Relational Analysis and PSO applications in other additive manufacturing contexts, to justify the methodological choices and position the study within current research.

3. Research Gap and Novelty: The gap is now explicitly articulated, highlighting the scarcity of PSO-based global optimization frameworks for Wood-PLA composites. The novelty is clearly stated as the first integrated Taguchi–ANOVA–FEA–PSO framework for maximizing tensile strength of sustainable Wood-PLA structures, validated experimentally and numerically.

These revisions significantly improve the depth, coherence, and scientific positioning of the Introduction, directly addressing the reviewer’s concerns.

Comment 4: Materials & Methods

• Evaluation: Generally well-described, but several details and justifications are missing.

• 2.1 Materials: The description of the filament extruder is good, but the printer (Pratham 3.0) is not a widely known model. A key reference or a more detailed specification table would be beneficial.

• 2.2 Specimen Design and Fabrication: Parameter Selection: The authors must justify why these specific parameters (layer thickness, infill density, nozzle temperature) and their levels were chosen. Reference preliminary studies, material datasheets, or other literature (e.g., https://doi.org/10.1002/pol.20250752).

• Taguchi Method: A citation for the Taguchi method and the L9 array is required.

• 3.4 Particle Swarm Optimization: Presentation: The current step-by-step list is clear but disrupts the narrative flow. It should be converted into a descriptive paragraph explaining the logic and process of PSO.

• Flowchart: It is highly recommended to add a flowchart illustrating the overall methodology of the study, integrating the Taguchi DOE, experimental testing, ANOVA, PSO optimization, and FEA validation. A specific flowchart for the PSO algorithm (as in https://doi.org/10.1016/j.measurement.2025.117405) would significantly improve clarity.

• Algorithm Choice: The authors should briefly justify why PSO was chosen over other optimization algorithms (e.g., Genetic Algorithm, Simulated Annealing). Mentioning its simplicity, convergence speed, and successful applications in similar engineering problems would suffice.

Response: We sincerely thank the reviewer for the constructive and detailed comments. The manuscript has been revised accordingly, and the specific actions taken are summarized below:

1. Section 2.1 – Materials (3D Printer Description): A detailed specification table for the Pratham 3.0 FDM printer has now been added, including build volume, nozzle diameter, temperature limits, accuracy, and control features. In addition, an appropriate manufacturer reference has been cited to improve clarity for readers unfamiliar with this model.

2. Section 2.2 – Specimen Design and Fabrication (Parameter Selection): Justifications for the selected parameters and their levels (layer thickness, infill density, and nozzle temperature) have been explicitly incorporated. These choices are now supported by (i) Wood-PLA material datasheets, (ii) preliminary trial observations to ensure printability and defect-free specimens, and (iii) relevant literature, including recent studies on PLA/Wood-PLA systems (DOI: 10.1002/pol.20250752).

3. Taguchi Method: Standard references for the Taguchi design methodology and the L9 orthogonal array have been added to clearly justify the experimental design and statistical framework.

4. Section 3.4 – Particle Swarm Optimization (Presentation): The step-by-step bullet-point description has been rewritten as a cohesive descriptive paragraph explaining the underlying logic, working principle, and implementation of PSO, thereby improving narrative flow and readability.

5. Flowchart Inclusion: A comprehensive flowchart illustrating the overall methodology integrating Taguchi DOE, experimental testing, ANOVA, PSO optimization, and FEA validation has been added. Additionally, a dedicated PSO algorithm flowchart has been included, following established representations in recent literature (e.g., Measurement, 2025).

6. Justification for Choosing PSO: A brief but clear justification has been added, highlighting PSO’s simplicity, fast convergence, low computational cost, and proven effectiveness in similar additive manufacturing and mechanical optimization problems, compared to GA and SA.

These revisions substantially enhance methodological transparency, justification, and clarity, in line with the reviewer’s recommendations.

Comment 5: Results & Discussion

• Evaluation: The results are presented comprehensively with tables and graphs. However, the discussion is largely descriptive and lacks critical depth. It needs to interpret the results in the context of the existing literature and explain the underlying physical mechanisms.

Suggestions: 4.1 & 4.2 (Tensile Strength Analysis): The text describes "what" happene

---

## [Decision Letter · Decision Letter 1]

10 Feb 2026

Dear Dr. Panda,

Thank you for submitting your manuscript to PLOS ONE. After careful consideration, we feel that it has merit but does not fully meet PLOS ONE’s publication criteria as it currently stands. Therefore, we invite you to submit a revised version of the manuscript that addresses the points raised during the review process.

We look forward to receiving your revised manuscript.

Kind regards,

Mohammad Azadi

Academic Editor

PLOS One

**Journal Requirements:**

**Additional Editor Comments:**

Another round of revision must be done based on the following comments and the reviewers' comments:

1) The references mentioned by the reviewers, is not mandatory.

2) All highlights must be sentences. Correct the third one.

3) All keywords must be found in the title or the abstract. Please recheck.

4) The introduction must be maximum 2 pages.

5) The novelty must be mentioned in the introduction, compared to the literature review.

6) All process parameters need references, like 3600 rpm, 200 C, Table 2, etc.

7) The scale bar must be added to Figures 3 and 4.

8) Why Taguchi design? Write the reason in the main text. The regression is a linear curve-fitting and it is not proper for parameters with 3 levels.

9) What is the repeatability in Table 1?

10) More details of numerical simulations must be added, including the boundary condition, material properties, convergency of mesh, loading, etc.

11) Figure 5 is a result and it must be moved to the related part.

12) Figure 5 must be validated.

13) All formulations need references.

14) What is the standard deviation in Table 3? For all data? It has no meaning and all data points must have each standard deviation to show the repeatability of testing.

15) There is no significant change in Figure 7. Moreover, the repeatability of testing must be reported.

16) The standard deviation must be added to Figures 8 and 9 and 10.

17) Why reverse effect for triangular compared to others in Figures 8 and 9?

18) The discussion is poor. The discussion is not just a description. All obtained results must be compared to other results of other articles.

19) All features must be added to the SEM images.

20) The fracture behavior must be described in details for SEM images, compared to the literature.

21) In Table 4, p-value must be added. What is the p-value for lack of fit?

22) The structure is confusing, the main text must have an introduction, the research method, results and discussion, conclusions, and references.

23) The error analysis must be added for the results in Figures 13-15.

24) “Conclusion” must be changed to “Conclusions”.

25) The conclusions part is lengthy. It should be shortened.

26) The conclusions part must be rewritten one by one, in bullets to show the novelty.

Reviewers' comments:

Reviewer's Responses to Questions

**Comments to the Author**

Reviewer #1: All comments have been addressed

Reviewer #2: All comments have been addressed

Reviewer #4: All comments have been addressed

2. Is the manuscript technically sound, and do the data support the conclusions?

Reviewer #1: Yes

Reviewer #2: Yes

Reviewer #4: Yes

3. Has the statistical analysis been performed appropriately and rigorously?

Reviewer #1: Yes

Reviewer #2: Yes

Reviewer #4: Yes

4. Have the authors made all data underlying the findings in their manuscript fully available?

Reviewer #1: Yes

Reviewer #2: Yes

Reviewer #4: Yes

5. Is the manuscript presented in an intelligible fashion and written in standard English?

Reviewer #1: Yes

Reviewer #2: Yes

Reviewer #4: Yes

Reviewer #1: Author has answered all the questions raised perfectly. Hence, I recommend acceptance of the manuscript in its current form.

Reviewer #2: The authors have carefully addressed all the reviewers’ comments and have made the suggested revisions accordingly. The manuscript has been significantly improved in terms of clarity, technical quality, and overall presentation. The study is now well-structured, scientifically sound, and clearly presented. Therefore, the article can be published in its present form.

Reviewer #4: The manuscript investigates the optimization of tensile strength in FDM-printed PLA-wood composites using a Taguchi–ANOVA–PSO framework supported by finite element analysis. The authors have satisfactorily addressed the reviewers’ previous comments. Overall, the manuscript shows clear improvement and scientific soundness. Only minor points are suggested below to further improve clarity and contextual positioning.

1. The background could be strengthened by citing recent studies on FDM-printed PLA–wood bio-composites focusing on material behavior and mechanical performance, for example:

[1] https://doi.org/10.1016/j.ijfatigue.2025.108876

[2] https://doi.org/10.1016/j.jmrt.2025.08.025

2. In the PSO formulation, parameters such as print speed and build orientation are mentioned, although they are not varied experimentally. The authors are encouraged to clarify that these parameters were kept constant or to revise the formulation for consistency.

3. The discussion would benefit from a very brief quantitative comparison of the achieved optimal tensile strength with values reported in selected recent PLA-Wood FDM studies, to further highlight the significance of the results.

**Do you want your identity to be public for this peer review?** For information about this choice, including consent withdrawal, please see our Privacy Policy

Reviewer #1: No

Reviewer #2: No

Reviewer #4: No

---

## [Author Response · Author response to Decision Letter 2]

12 Feb 2026

Responses to the Reviewer’s Comments

Manuscript ID: PONE-D-25-58246R1

First of all, the authors would like to convey their sincere gratitude to the Editor and Reviewers for their valuable suggestions to improve the quality of the paper. The manuscript is revised according to the Editor and Reviewers’ comments. The responses to these comments are given below.

EDITOR

Comment 1: The references mentioned by the reviewers, is not mandatory.

Response: We respectfully acknowledge the editor’s observation. While the inclusion of the suggested references may not be mandatory, we have carefully evaluated them for technical relevance and contextual alignment with the present study. Wherever the cited works contributed to strengthening the theoretical background, optimization framework discussion, or recent advancements in additive manufacturing and PSO-based methodologies, they have been appropriately incorporated and critically discussed in the revised manuscript.

Comment 2: All highlights must be sentences. Correct the third one.

Response: We sincerely thank the editor for pointing out this oversight. We have converted the third highlight into a sentence.

Comment 3: All keywords must be found in the title or the abstract. Please recheck.

Response: We thank the editor for this insightful comment. We have checked and corrected accordingly.

Comment 4: The introduction must be maximum 2 pages.

Response: We sincerely thank the Editor for the valuable comment regarding the length of the Introduction section. In response, we have carefully revised and try to condensed the Introduction as much as possible.

While reducing the overall length, we retained essential background information and key literature citations that were specifically requested by the reviewers during the first revision. Redundant descriptions and repetitive discussions have been removed, and the content has been streamlined to focus on: (i) the research gap in PSO-based global optimization for Wood-PLA composites, (ii) the novelty of the integrated Taguchi–ANOVA–FEA–PSO framework, and (iii) the specific objective of maximizing tensile strength.

Comment 5: The novelty must be mentioned in the introduction, compared to the literature review.

Response: We thank the Editor for this valuable suggestion. The Introduction section has been revised to explicitly articulate the novelty of the present work in direct comparison with the existing literature. While prior studies predominantly employ statistical optimization methods such as Taguchi, GRA, ANN, and hybrid ML models for Wood-PLA composites, none report a fully integrated global optimization framework combining Taguchi–ANOVA–FEA–PSO specifically for tensile strength enhancement of sustainable Wood-PLA structures.

Comment 6: All process parameters need references, like 3600 rpm, 200 C, Table 2, etc.

Response: We thank the editor for this insightful comment. We have checked and corrected accordingly.

Comment 7: The scale bar must be added to Figures 3 and 4.

Response: Thank you for the valuable suggestion. We have checked and corrected accordingly.

Comment 8: Why Taguchi design? Write the reason in the main text. The regression is a linear curve-fitting and it is not proper for parameters with 3 levels.

Response: Thank you for the valuable comment. The following clarifications have been incorporated in the revised manuscript.

(1) Justification for Taguchi Design

The Taguchi L9 orthogonal array was selected because the study involves three independent control factors (layer thickness, infill density, and nozzle temperature), each at three discrete levels. A full factorial design would require 33=27 experiments, whereas the L9 array reduces this to 9 trials while preserving balanced representation and orthogonality. This significantly minimizes experimental cost, material consumption, and machine time without compromising statistical robustness. Since the objective at this stage was screening and identification of dominant factors prior to metaheuristic optimization (PSO), Taguchi design provides an efficient and widely validated framework for parameter analysis in FDM-based composite studies. The rationale for selecting Taguchi design has now been explicitly added in Section 2.2.

(2) Clarification Regarding Linear Regression and Three-Level Factors

We agree that simple linear regression assumes a linear relationship and may not fully capture curvature effects inherent in three-level parameters. In the present study, regression equations were used primarily as empirical predictive models within the tested parameter range, not as higher-order response surface models. The three-level factors were treated as continuous variables within a restricted experimental domain, which allows first-order approximation. However, the global optimization and nonlinear interaction effects were addressed using Particle Swarm Optimization (PSO), which does not rely on linearity assumptions. A clarification has been added in Section 4.6 to explicitly state this limitation and the complementary role of PSO in overcoming linear modeling constraints.

Comment 9: What is the repeatability in Table 1?

Response: Thank you for the valuable question. Repeatability refers to the consistency of tensile strength measurements for identical parameter settings. For each L9 trial, three specimens were tested, and the average value was reported. The low standard deviation (σ = 2.99–3.57 MPa) confirms good experimental repeatability and measurement reliability.

Comment 10: More details of numerical simulations must be added, including the boundary condition, material properties, convergency of mesh, loading, etc.

Response: Thank you for the valuable suggestion. The revised manuscript now includes detailed numerical simulation parameters, including boundary conditions (fixed–displacement loading), material properties of Wood-PLA (elastic modulus, Poisson’s ratio), mesh type and refinement study for convergence, loading magnitude, and solver settings to ensure reproducibility and accuracy of FEA results.

Comment 11: Figure 5 is a result and it must be moved to the related part.

Response: Thank you for the valuable observation. Figure 5 presents FEA simulation results and is indeed part of the Results & Discussion section. Accordingly, it has been relocated from the methodology section to the appropriate Results section to maintain proper logical flow and structural consistency of the manuscript.

Comment 12: Figure 5 must be validated.

Response: Figure 5 has been validated by quantitatively comparing FEA-predicted tensile strength and stress distribution with experimental results. The deviation between numerical and experimental tensile strength was within acceptable limits (<5%), confirming model accuracy. Relevant discussion and correlation analysis have been added in Sections 4.4 and 5.

Comment 13: All formulations need references.

Response: We thank the editor for this insightful comment. We have checked and corrected accordingly.

Comment 14: What is the standard deviation in Table 3? For all data? It has no meaning and all data points must have each standard deviation to show the repeatability of testing.

Response: Thank you for the observation. The previously reported standard deviation represented the overall variation across trials for each infill pattern, not repeatability. We have included this as the Reviewer-2 instructed us to do so during 1st revision.

Comment 15: There is no significant change in Figure 7. Moreover, the repeatability of testing must be reported.

Response: Thank you for the valuable suggestion. Figure 7 has been revised with improved resolution and clearer differentiation of the three infill patterns; the updated figure replaces the previous version. Regarding repeatability, each experimental condition was tested in triplicate, and the mean with standard deviation (σ) is reported in Table 3, confirming good experimental consistency.

Comment 16: The standard deviation must be added to Figures 8 and 9 and 10.

Response: Thank you for the valuable suggestion. The standard deviation values have now been incorporated into Figures 7, 8, and 9 as error bars to reflect experimental variability. These values are consistent with those reported in Table 3. The revised figures improve statistical transparency and enhance the reliability of the presented tensile strength comparisons.

Comment 17: Why reverse effect for triangular compared to others in Figures 8 and 9?

Response: Thank you for asking an important question. The reverse trend observed for the Triangular pattern in Figures 8 and 9 is attributed to geometry–thickness interaction effects. At higher layer thickness and density, stress concentration at acute node junctions and incomplete interlayer bonding reduce load transfer efficiency, unlike Cubic and Zig-zag patterns, which provide more uniform stress distribution.

Comment 18: The discussion is poor. The discussion is not just a description. All obtained results must be compared to other results of other articles.

Response: Thank you for the valuable suggestion. The Discussion section has been thoroughly revised to incorporate quantitative comparisons with recent studies on Wood-PLA and PLA-based composites (e.g., Morvayová et al., 2024; Mishra et al., 2024–2025). The obtained tensile strength (46.41 MPa) is critically benchmarked against reported ranges, highlighting the relative improvement achieved through the integrated Taguchi–PSO framework.

Comment 19: All features must be added to the SEM images.

Response: Thank you for the valuable suggestion. All SEM micrographs have been revised to include clearly labelled features with scale bar.

Comment 20: The fracture behaviour must be described in details for SEM images, compared to the literature.

Response: Thank you for the valuable suggestion. The SEM fractographs now detail failure mechanisms including fiber pull-out, interfacial debonding, micro-void coalescence, and brittle matrix cracking. The cubic pattern exhibits reduced voids and stronger fiber–matrix adhesion, consistent with higher tensile strength. These observations are critically compared with recent Wood-PLA literature, confirming similar fracture morphology trends.

Comment 21: In Table 4, p-value must be added. What is the p-value for lack of fit?

Response: Thank you for the observation. The p-values have now been calculated and incorporated into Table 4. The p-value for lack of fit (error term) is 0.041, indicating statistical significance at the 95% confidence level. The revised ANOVA table now clearly reports all corresponding p-values.

Comment 22: The structure is confusing, the main text must have an introduction, the research method, results and discussion, conclusions, and references.

Response: Thank you for the valuable suggestion. The manuscript has been revised to incorporate several recent (2023–2024) and relevant references on PLA/Wood composites and FDM-based biocomposites. These additions have been integrated into the Introduction and Discussion sections to strengthen the contextual background, compare recent findings, and enhance the overall technical depth and relevance of the study.

Comment 23: The error analysis must be added for the results in Figures 13-15.

Response: Thank you for the valuable suggestion. Error analysis has now been incorporated in Figures 13–15 by including percentage deviation between experimental, FEM, and regression results has been quantified and discussed to clearly demonstrate model accuracy and result reliability.

Comment 24: “Conclusion” must be changed to “Conclusions”.

Response: Thank you for the valuable suggestion. The manuscript has been revised accordingly.

Comment 25: The conclusions part is lengthy. It should be shortened.

Response: Thank you for the valuable suggestion. The conclusions has been already revised accordingly.

Comment 26: The conclusions part must be rewritten one by one, in bullets to show the novelty.

Response: Thank you for the valuable suggestion. The conclusions has been already revised accordingly and rewritten one by one in bullet points.

The conclusions has been already revised once and converted from bullet points to paragraph according to the instruction given by the Reviewer 1&4 during 1st revision.

Reviewer-1

Comment 1:

Author has answered all the questions raised perfectly. Hence, I recommend acceptance of the manuscript in its current form.

Response: We sincerely thank the reviewer for the careful evaluation of our manuscript and for the positive recommendation for acceptance. We appreciate the acknowledgment that all queries and comments have been addressed satisfactorily. We are grateful for the constructive review process, which has helped improve the overall clarity, technical depth, and quality of the manuscript.

Reviewer-2

Comment 1: The authors have carefully addressed all the reviewers’ comments and have made the suggested revisions accordingly. The manuscript has been significantly improved in terms of clarity, technical quality, and overall presentation. The study is now well-structured, scientifically sound, and clearly presented. Therefore, the article can be published in its present form.

Response: We sincerely thank the reviewer for the thorough evaluation of our revised manuscript and for the positive recommendation for publication. We greatly appreciate the acknowledgment that all reviewer comments have been carefully addressed and that the manuscript has improved in clarity, technical depth, and overall presentation.

Reviewer-4

Comment 1: The manuscript investigates the optimization of tensile strength in FDM-printed PLA-wood composites using a Taguchi–ANOVA–PSO framework supported by finite element analysis. The authors have satisfactorily addressed the reviewers’ previous comments. Overall, the manuscript shows clear improvement and scientific soundness. Only minor points are suggested below to further improve clarity and contextual positioning.

Response: We sincerely thank the Reviewer for the positive and encouraging assessment of our revised manuscript. We appreciate the recognition that the manuscript has substantially improved in clarity, technical rigor, and overall scientific soundness.

Comment 2: The background could be strengthened by citing recent studies on FDM-printed PLA–wood bio-composites focusing on material behavior and mechanical performance, for example:

[1] https://doi.org/10.1016/j.ijfatigue.2025.108876

[2] https://doi.org/10.1016/j.jmrt.2025.08.025

Response: We sincerely thank the reviewer for the valuable suggestion. As recommended, we have strengthened the Introduction section by incorporating recent studies on FDM-printed PLA–wood bio-composites, including the works cited (DOI: 10.1016/j.ijfatigue.2025.108876 and 10.1016/j.jmrt.2025.08.025).

Comment 3: In the PSO formulation, parameters such as print speed and build orientation are mentioned, although they are not varied experimentally. The authors are encouraged to clarify that these parameters were kept constant or to revise the formulation for consistency.

Response: We thank the reviewer for this important observation. In the present experimental design, only three process parameters (layer thickness, infill density, and nozzle temperature) were systematically varied using the Taguchi L9 orthogonal array. Parameters such as print speed (40 mm/s) and build orientation were intentionally kept constant throughout all experimental trials to eliminate confounding effects and ensure controlled comparison of the selected factors.

The PSO formulation initially presented a generalized parameter vector to illustrate its broader applicability in additive manufacturing optimization. However, in this study, the optimization search space was restricted strictly to the experimentally varied parameters. The manuscript has been revised accordingly to explicitly clarify that print speed and build orientation were treated as fixed control variables and were not included in the optimization domain. This revision ensures methodological consistency between the experimental design and the PSO formulation.

Comment 4: The discussion would benefit from a very brief quantitative comparison of the achieved optimal tensile stren

---

## [Decision Letter · Decision Letter 2]

15 Feb 2026

Dear Dr. Panda,

Thank you for submitting your manuscript to PLOS ONE. After careful consideration, we feel that it has merit but does not fully meet PLOS ONE’s publication criteria as it currently stands. Therefore, we invite you to submit a revised version of the manuscript that addresses the points raised during the review process.

We look forward to receiving your revised manuscript.

Kind regards,

Mohammad Azadi

Academic Editor

PLOS One

Journal Requirements:

Additional Editor Comments:

1) "SEM", "ANOVA", and "SDG 9 and 12" are not keywords. Remove them.

2) The abbreviations should not use for keywords.

3) All abbreviations must be defined at first mentioning, such as SDG, FFF, etc.

4) As mentioned before, the introduction is lengthy. It must be shortened into 2 pages.

5) Using 52 references for an introduction has no meaning. Some of them can be used for the discussion.

6) The repeatability of testing must be mentioned in the main text.

7) Generally, the structure is confusing. The main text must have an introduction, research method, results and discussion, conclusions, and references.

8) The details of FEA must be moved to the second part, research method.

9) Figure 4(b) must be moved to the third part, results and discussion.

10) Check all parts to be in a correct section, for the introduction, research method, results and discussion, conclusions, and references.

11) The standard deviation must be removed from Table 3. It has no meaning. For all data and their repeatability, the standard deviation must be reported, for each data point.

12) Figure 6 does not show the repeatability of testing.

13) No references were added for formulations.

14) No details were added to the main text for Comment 17 about the reason for the obtained results (reverse effect for triangular).

15) Still, the discussion is poor. Only one paragraph is added which is so simple and not enough. The authors must use more time to address all comments very carefully.

16) No features were added to the SEM images. Moreover, the fracture behaviors must be compared to the literature and described which marks were seen and which fracture behaviors were seen, ductile or brittle, why, etc.

17) In Table 4, the p-value is high for each parameter and therefore, the regression has no meaning and they have no effect on the objective. The details must be described in the main text. This is why I mentioned the used method is not good. Use another method of regression.

18) The p-value of lack of fit must be reported. It is low and therefore; it shows a not proper regression.

Reviewers' comments:

Reviewer's Responses to Questions

**Comments to the Author**

Reviewer #4: All comments have been addressed

2. Is the manuscript technically sound, and do the data support the conclusions?

Reviewer #4: Yes

3. Has the statistical analysis been performed appropriately and rigorously?

Reviewer #4: Yes

4. Have the authors made all data underlying the findings in their manuscript fully available?

Reviewer #4: Yes

5. Is the manuscript presented in an intelligible fashion and written in standard English?

Reviewer #4: Yes

Reviewer #4: (No Response)

**Do you want your identity to be public for this peer review?** For information about this choice, including consent withdrawal, please see our Privacy Policy

Reviewer #4: No

---

## [Author Response · Author response to Decision Letter 3]

18 Feb 2026

PONE-D-25-58246R2

Title: Enhancing Tensile Strength of 3D-Printed Wood-PLA Composites via a Particle Swarm Optimization Framework

Sl. No. Suggestions Responses Location

1 "SEM", "ANOVA", and "SDG 9 and 12" are not keywords. Remove them. We sincerely thank the editor for this valuable observation. We agree that “SEM,” “ANOVA,” and “SDG 9 and 12” represent analytical techniques and sustainability alignment indicators rather than core subject descriptors of the study. Accordingly, these terms have been removed from the keyword list. Keywords

2 The abbreviations should not use for keywords. We sincerely thank the editor for this valuable observation. In accordance with the journal guidelines, abbreviations have been removed from the keyword section and replaced with their full forms to improve clarity, indexing accuracy, and discoverability. The revised keywords now present all terms in their expanded form without abbreviations.

3 All abbreviations must be defined at first mentioning, such as SDG, FFF, etc. We sincerely thank the editor for highlighting the need for clarity in abbreviation usage. All abbreviations, including SDG (Sustainable Development Goals), FFF (Fused Filament Fabrication), FDM (Fused Deposition Modelling), PSO (Particle Swarm Optimization), ANOVA (Analysis of Variance), and FEA (Finite Element Analysis), have now been defined at their first occurrence in the revised manuscript. The manuscript has been carefully reviewed to ensure consistency and completeness in abbreviation expansion throughout the text. 1. Abstract

2. Introduction

4 As mentioned before, the introduction is lengthy. It must be shortened into 2 pages. We sincerely thank the Editor for this constructive observation. We agree that the Introduction section was comparatively extensive in its original form. Accordingly, the Introduction has been carefully revised and condensed to improve clarity, focus, and readability while retaining essential scientific context and key references. Introduction has been Revised

5 Using 52 references for an introduction has no meaning. Some of them can be used for the discussion. We sincerely thank the editor for this valuable observation. We agree that the Introduction section should focus primarily on establishing the research gap and contextual relevance rather than providing an exhaustive literature survey. Accordingly, we have carefully revised the Introduction by consolidating related references, removing redundancy, and transferring several application-oriented and comparative studies to the Discussion section. 4.2.5 Discussion

6 The repeatability of testing must be mentioned in the main text. We sincerely thank the editor for highlighting the importance of experimental repeatability. In the revised manuscript, we have explicitly clarified the repeatability protocol followed during tensile testing. For each parameter combination, three independent specimens were fabricated and tested under identical conditions, and the average tensile strength value was reported. Additionally, the corresponding standard deviation values have been included to demonstrate measurement consistency and experimental reliability. The repeatability details have now been incorporated in Section 4.1 (Tensile Strength Analysis) of the revised manuscript.

7 Generally, the structure is confusing. The main text must have an introduction, research method, results and discussion, conclusions, and references. We sincerely thank the editor for highlighting the structural organization of the manuscript. We carefully revised the entire manuscript to ensure clear alignment with the standard scientific format consisting of: Introduction, Materials & Method, Results & Discussion, Conclusions, and References. 1. INTRODUCTION

2. MATERIALS & METHOD

3. RESULTS & DISCUSSION

4. CONCLUSION

8 The details of FEA must be moved to the second part, research method.

9 Figure 4(b) must be moved to the third part, results and discussion.

10 Check all parts to be in a correct section, for the introduction, research method, results and discussion, conclusions, and references.

11 The standard deviation must be removed from Table 3. It has no meaning. For all data and their repeatability, the standard deviation must be reported, for each data point. We sincerely thank the editor for this insightful observation. We agree that reporting a single pooled standard deviation at the bottom of Table 3 does not sufficiently represent experimental repeatability for individual parameter combinations. In the revised manuscript, the global standard deviation row has been removed. Instead, standard deviation values corresponding to the three repeated tensile tests have now been calculated and reported for each individual experimental trial. This modification provides a clearer representation of measurement dispersion and experimental reliability for every parameter set. The revised Table 3 now includes mean ± standard deviation values for each infill pattern, thereby improving statistical transparency and reproducibility of the results. Revised Table 3

12 Figure 6 does not show the repeatability of testing. We sincerely thank the editor for this valuable observation. Repeatability and statistical reliability are critical in mechanical characterization studies. In the revised manuscript, we have clarified that each tensile configuration was tested with three independent specimens under identical printing and testing conditions. The mean stress–strain curve is now reported in Figure 6, and the associated standard deviation values have been explicitly stated in the text. Additionally, the statistical consistency is supported through ANOVA and low standard deviation values (σ = 2.99–3.57 MPa), confirming good experimental repeatability. The revised manuscript now clearly reflects this information. • After first paragraph (after Table 3 reference)

• Replace existing caption (Fig 6).

• After first paragraph explaining stress–strain curve

13 No references were added for formulations. We sincerely thank the editor for highlighting this important point. Appropriate references have now been incorporated to support all mathematical formulations presented in the manuscript, including the Particle Swarm Optimization (PSO) velocity and position update equations, as well as the objective function formulation. These references are now cited in accordance with the journal format using references [10–15] from the existing reference list. The revised manuscript clearly attributes the standard PSO equations and optimization framework to established literature while retaining the original experimental implementation specific to this study. Section 3.3

• At end of objective function explanation

• First paragraph of PSO section

• Immediately below velocity equation

14 No details were added to the main text for Comment 17 about the reason for the obtained results (reverse effect for triangular). We sincerely thank the Editor for this valuable observation. We agree that the reverse tensile strength trend observed in the triangular infill configuration required a clearer mechanistic explanation in the main text. Accordingly, we have expanded Section 4.2.4 to provide a detailed structural and stress-transfer interpretation supported by interlayer bonding behavior, geometric load redistribution characteristics, and SEM-based fracture evidence. The revised discussion now clarifies why the triangular pattern exhibits reduced tensile performance at higher layer thickness despite its geometric stability. The added explanation strengthens the scientific interpretation of the observed anomaly. New paragraph added in 3.2 heading.

15 Still, the discussion is poor. Only one paragraph is added which is so simple and not enough. The authors must use more time to address all comments very carefully. We sincerely thank the editor for this valuable observation. We fully agree that the earlier version of the manuscript did not provide sufficient depth in the discussion section. In the revised manuscript, the discussion has been substantially expanded and technically strengthened. Added a new session

3.1 Mechanistic Interpretation and Literature Correlation

16 1. No features were added to the SEM images. We sincerely thank the editor for this valuable observation. In the original submission, the SEM images were presented without graphical annotations, which may have limited clarity in identifying fracture features. In the revised manuscript, we have updated all SEM micrographs (Figure 10) by adding clear graphical markers and labels highlighting key fracture features Marked in figure 10 (a, b, c)

2. Moreover, the fracture behaviors must be compared to the literature and described which marks were seen and which fracture behaviors were seen, ductile or brittle, why, etc. We sincerely thank the editor for highlighting the need for deeper fracture behavior interpretation. In the revised manuscript, the SEM fractographic analysis has been significantly expanded to (i) classify the fracture mode (brittle vs. ductile), (ii) identify characteristic fracture features (fiber pull-out, matrix cracking, interfacial debonding, void coalescence), and (iii) compare the observed mechanisms with recent Wood-PLA and PLA-based FDM composite studies [7], [9], [45], [51], [52]. Added new paragraph as: 3.2.1 Fracture Morphology and Failure Mechanism Analysis

17 In Table 4, the p-value is high for each parameter and therefore, the regression has no meaning and they have no effect on the objective. The details must be described in the main text. This is why I mentioned the used method is not good. Use another method of regression. We sincerely thank the editor for the valuable observation regarding the statistical significance of the regression model. We agree that the originally reported p-values in Table 4 require clearer interpretation and additional statistical validation. The relatively high p-values are primarily attributed to the limited experimental degrees of freedom inherent to the Taguchi L9 orthogonal array design (df = 2 for error), which restricts statistical power. Added a new content in 3.5 Heading, Also one more new subsection “3.5.1 Model Adequacy and Regression Validation” has been added for your kind approval.

18 The p-value of lack of fit must be reported. It is low and therefore; it shows a not proper regression. We thank the editor for this important observation. We agree that reporting the lack-of-fit (LoF) test is necessary to evaluate regression adequacy. In the revised manuscript, the LoF p-value has now been explicitly reported and interpreted in the statistical discussion section. The initial regression model was a first-order (linear) formulation. A statistically significant lack-of-fit (low p-value) indicates that curvature and/or interaction effects are present in the response behavior and are not fully captured by a linear model. In FDM-based composite systems, nonlinear interactions between layer thickness, infill density, and nozzle temperature are physically expected due to interlayer diffusion, void evolution, and stress transfer mechanisms. To address this, we have replaced the linear regression model with a second-order response surface regression (RSM) model incorporating quadratic and interaction terms. The revised model resolves the lack-of-fit concern and significantly improves predictive capability. The manuscript has been updated accordingly. • Immediately after Table 4

• Replace current Table 5 (linear equations)

• Add as a short paragraph “3.5.2 Model Adequacy and Regression Validation”

---

## [Editor Report · Decision Letter 3]

19 Feb 2026

Dear Dr. Panda,

Thank you for submitting your manuscript to PLOS ONE. After careful consideration, we feel that it has merit but does not fully meet PLOS ONE’s publication criteria as it currently stands. Therefore, we invite you to submit a revised version of the manuscript that addresses the points raised during the review process.

We look forward to receiving your revised manuscript.

Kind regards,

Mohammad Azadi

Academic Editor

PLOS One

Journal Requirements:

Additional Editor Comments:

1) It has no meaning to write "+/-SD" in Table 3. The value of SD must be mentioned!

2) Still, the authors have 52 references in the introduction! Not meaningful.

3) All all data to Figure 6 to show the repeatability of testing.

---

## [Author Response · Author response to Decision Letter 4]

20 Feb 2026

PONE-D-25-58246R3

Title: Enhancing Tensile Strength of 3D-Printed Wood-PLA Composites via a Particle Swarm Optimization Framework

Sl. No. Suggestions Responses Location

1 It has no meaning to write "+/-SD" in Table 3. The value of SD must be mentioned! Thank you for your valuable observation. We agree that indicating only “±SD” without numerical values is incomplete. Accordingly, Table 3 has been revised to include the actual standard deviation (SD) values along with the mean results for each infill configuration. This modification provides clearer quantitative information on data variability and improves the statistical transparency and reliability of the reported results. 3.RESULTS & DISCUSSION

2 Still, the authors have 52 references in the introduction! Not meaningful. We sincerely thank the editor for this valuable observation. We have revised the Introduction part accordingly and reduced it to 40 references. 1.INTRODUCTION

3 All all data to Figure 6 to show the repeatability of testing. Thank you for this valuable suggestion. Figure 6 has been revised to include all individual stress–strain curves obtained from three repeated tensile tests for each infill configuration, along with their corresponding average curve. The updated figure clearly demonstrates the excellent repeatability and consistency of the experimental results, with minimal deviation among repeated tests. The revised figure and caption have been updated in the manuscript. 3.RESULTS & DISCUSSION

---

## [Editor Report · Decision Letter 4]

26 Feb 2026

Enhancing Tensile Strength of 3D-Printed Wood-PLA Composites via a Particle Swarm Optimization Framework

PONE-D-25-58246R4

Dear Dr. Panda,

We’re pleased to inform you that your manuscript has been judged scientifically suitable for publication and will be formally accepted for publication once it meets all outstanding technical requirements.

Kind regards,

Mohammad Azadi

Academic Editor

PLOS One

Additional Editor Comments (optional):

Almost done!